# Self-Supervised Set Representation Learning for Unsupervised Meta-Learning

**Dong Bok Lee**[1][*]    **Seanie Lee**[1][*]    **Kenji Kawaguchi**[2]    **Yunji Kim**[3]    **Jihwan Bang**[3]
**Jung-Woo Ha**[3]    **Sung Ju Hwang**[1]
KAIST[1], National University of Singapore[2], NAVER[3]
{markhi, lsnfamily02, sjhwang82}@kaist.ac.kr
{yunji.kim, jihwan.bang, jungwoo.ha}@navercorp.com
kenji@comp.nus.edu.sg

## Abstract

*Unsupervised meta-learning* (UML) essentially shares the spirit of self-supervised learning (SSL) in that their goal aims at learning models without any human supervision so that the models can be adapted to downstream tasks. Further, the learning objective of self-supervised learning, which pulls positive pairs closer and repels negative pairs, also resembles metric-based meta-learning. Metric-based meta-learning is one of the most successful meta-learning methods, which learns to minimize the distance between representations from the same class. One notable aspect of metric-based meta-learning, however, is that it is widely interpreted as a *set-level* problem since the inference of discriminative class prototypes (or set representations) from few examples is crucial for the performance of downstream tasks. Motivated by this, we propose Set-SimCLR, a novel self-supervised set representation learning framework for targeting UML problem. Specifically, our Set-SimCLR learns a set encoder on top of instance representations to maximize the agreement between two sets of augmented samples, which are generated by applying stochastic augmentations to a given image. We theoretically analyze how our proposed set representation learning can potentially improve the generalization performance at the meta-test. We also empirically validate its effectiveness on various benchmark datasets, showing that Set-SimCLR largely outperforms both UML and instance-level self-supervised learning baselines.

## 1 Introduction

One of the most challenging and long-standing problems in machine learning is *unsupervised learning* which aims at learning generalizable representations without human supervision, which can be transferred to diverse downstream tasks. Meta-learning (Finn et al., 2017; Snell et al., 2017) is a popular framework for learning models that quickly adapt to novel tasks on the fly with few examples, and thus shares the spirit of unsupervised learning in that it seeks more efficient and effective learning procedures than learning from scratch. However, the essential difference between unsupervised learning and meta-learning is that most meta-learning approaches have been built on a supervised learning scheme and require human-crafted task distributions. In order to tackle this limitation, several previous works (Hsu et al., 2019; Khodadadeh et al., 2019; 2021; Lee et al., 2021) have proposed *unsupervised meta-learning* (UML) frameworks which combine unsupervised learning and meta-learning. They train a model with unlabeled data such that the model can adapt to unseen tasks with few labels.

Meanwhile, self-supervised learning (Chen et al., 2020a;b; He et al., 2020; Chen et al., 2020c; 2021; Grill et al., 2020; Zbontar et al., 2021) (SSL) is rising as a promising learning paradigm to learn transferable representations from unlabeled data in a *task-agnostic* manner. These methods rely on pretext tasks generated from data, and a popular pretext task is to maximize the agreement between different views of the same image in the latent space. The different views are easily obtained by sequentially applying pre-defined stochastic augmentations to an image. The main applications of these SSL methods essentially resemble the problem scenarios of UML, where we aim to transfer the learned representations to various downstream tasks. Further, the learning objective of SSL is also closely related to metric-based meta-learning (Ni et al., 2022), which is one of the most

---
[*]Equal contribution

successful meta-learning methods. Metric-based meta-learning (Snell et al., 2017) learns to minimize the distance between representations from the same class, while SSL pulls positive pairs closer and repels negative pairs. This motivates us to design a SSL method for addressing the UML problem.

Most SSL methods have focused on learning meaningful instance visual features. The importance of the instance feature is clear for generalization to unseen tasks coming with few examples, however, a meta-learning problem is often interpreted as a *set*-level problem in the literature of metric-based meta-learning. It has been widely shown that inference of discriminative class prototypes (or set representations) from few examples is crucial for the performance of downstream tasks. For example, Snell et al. (2017) basically takes an average of features belonging to the same class as a prototype (set representation). Similarly, Gordon et al. (2019); Iakovleva et al. (2020) propose Bayesian framework to learn stochastic prototypes using multi-layer perceptron and properly reflect uncertainty originating from few examples. Further, Triantafillou et al. (2019) propose to fine-tune the prototype with supervised loss. Inspired by the successes of set representation in few-shot learning, we propose a self-supervised set representation learning framework for UML.

The underlying assumption of SSL is that two different views of an image share most visual semantics. Built upon this idea, we construct two sets where each set consists of different views of the same image and maximize the agreement between them. Concretely, we repeatedly apply stochastic augmentations to each image of the mini-batch multiple times and construct a set consisting of the augmented images. Then we divide the set by half into two sets which are considered to be a positive pair of sets. Given a positive set pair, similar to Chen et al. (2020a), the other sets within mini-batch are considered as negative sets. We use attention-based set encoder (Vaswani et al., 2017; Lee et al., 2019) to obtain set representations. The set encoder is trained to reduce the distance of positive sets and increase that of negative sets. We dub our framework Set-SimCLR. At meta-test, we initialize each row of the weight for a linear classifier with the learned representation of the set composed of instances belonging to the same class, and the classifier is then optimized with supervised loss.

We motivate our algorithmic design of Set-SimCLR based on theoretical analysis. Specifically, we study how our set representation can potentially improve the final performance and the reason why we use set representations as the initialization of classifier weights. We then empirically validate our Set-SimCLR by comparing it against four UML methods and four instance-level SSL methods. We find that our method outperforms the baselines on six benchmark datasets, including Mini-ImageNet (Ravi & Larochelle, 2017), Tiny-ImageNet (Le & Yang, 2015), CIFAR100 (Krizhevsky et al., 2009), Aircraft (Maji et al., 2013), Stanford Cars (Krause et al., 2013) and CUB (Wah et al., 2011) datasets.

We summarize our contributions as follows:

- We introduce Set-SimCLR framework for solving unsupervised meta-learning problem, which learns both instance and set representations for downstream tasks.
- We provide a theoretical motivation of Set-SimCLR and study how the set representation potentially improves few-shot classification performance.
- The proposed Set-SimCLR outperforms the previous UML baselines and self-supervised learning baselines by significant margins in all the datasets we consider.

## 2 RELATED WORK

**Unsupervised Meta-Learning (UML)**  To tackle the limitation of supervised meta-learning, several UML works have been proposed to construct pseudo-tasks for meta-training by clustering data on an unsupervised embedding space (Hsu et al., 2019), data augmentation (Khodadadeh et al., 2019), or harvesting synthetic data from the latent space of generative models (Khodadadeh et al., 2021). Contrary to the works focusing on generating pseudo tasks, Meta-GMVAE (Lee et al., 2021) introduces a Mixture of Gaussian priors by performing Expectation-Maximization during the meta-training and the meta-test. To our knowledge, none of the existing works have proposed to tackle UML with self-supervised set representation learning, although Lee et al. (2021); Ericsson et al. (2021) use a backbone network pretrained with instance-level SSL objective.

**Set Representation**  DeepSets (Zaheer et al., 2017) independently processes elements and aggregates them with either min, max, mean or sum operation to obtain permutation invariant set encoding. To tackle the lack of expressiveness of Deepsets, Set Transformer (Lee et al., 2019) utilize self-attention to model the pairwise interaction of elements in a set. Instead of designing a more

expressive neural architecture for set encoding, several methods are proposed to learn set representation by minimizing the distance between an input set and a trainable reference set using a bipartite matching (Skianis et al., 2020), an optimal transport (Mialon et al., 2020; dan Guo et al., 2022), or Wasserstein embedding (Kolouri et al., 2020). Note that our self-supervised set representation learning framework is agnostic to any set encoding and any of them can be utilized for ours.

**Self-supervised Learning** Recently, a large volume of works has proposed self-supervised learning methods. The core idea is the representation of differently augmented views of the same image should be similar. Note that we introduce just a few of them which we consider as baselines in our experiments. SimCLR (Chen et al., 2020a;b) is one of the most representative contrastive frameworks where two views of the same image are pulled together while the negative pairs are repulsed. MOCO (He et al., 2020; Chen et al., 2020c; 2021) builds a dynamic feature dictionary using a queue and momentum encoder and learns to minimize contrastive loss from the dictionary. Meanwhile, several works show that non-contrastive approaches can learn meaningful representation without a latent feature collapse. For example, BYOL (Grill et al., 2020) leverages two identical networks where one of them is a momentum encoder to encode different views of images and minimizes the distance between positive pairs. Barlow Twins (Zbontar et al., 2021) computes a cross-correlation matrix between a different view of images and optimize it to be close to an identity matrix. Recently, MAE (He et al., 2022) masks an image and reconstructs the masked input to learn a meaningful representation of images. In this work, we exploit the effectiveness of self-supervised learning on UML, especially combined with our proposed set representation learning.

## 3 METHOD

In this section, we describe problem setting of unsupervised meta-learning (UML) and self-supervised set representation learning, Set-SimCLR. We depict an overview of our method in Figure 1.

### 3.1 PROBLEM STATEMENT

For UML problem, we can only access to an unlabeled dataset $\mathcal{D}^u = \{\mathbf{x}_i\}_{i=1}^U$ for meta-training. Same as most existing meta-learning works (Finn et al., 2017; Snell et al., 2017), we assume meta-test data follows the same data distribution of unlabeled dataset $\mathcal{D}^u$ while having a different set of classes. At meta-test time, we are given a set of $N$-way $S$-shot classification tasks and each task consists of a support set $\mathcal{D}^s = \{(\mathbf{x}_i^s, y_i^s)\}_{i=1}^{N \times S}$, and a query set $\mathcal{D}^q = \{\mathbf{x}_i^q\}_{i=1}^{N \times Q}$. The final goal is to leverage the model trained on the unlabeled data to predict labels of the query set with the help of the support set.

### 3.2 SELF-SUPERVISED CONTRASTIVE LEARNING

Before introducing our method, we first describe one of the most successful self-supervised learning methods SimCLR (Chen et al., 2020a). SimCLR is a contrastive learning framework that maximizes agreement between differently augmented views of the same instance in the latent space. Specifically, it first randomly samples a mini-batch of $M$ images $\{\mathbf{x}_m\}_{m=1}^M$ and obtains two different views of each image using stochastic data augmentation, resulting in $2M$ instances $\{(\mathbf{x}_{m,1}, \mathbf{x}_{m,2})\}_{m=1}^M$. There are two components: 1) a base encoder $f$ extracting feature representations and 2) a projection head $g$ mapping the representation to the latent space where the contrastive loss is applied. With the encoder and projection head, the latent representation of each image is obtained as $\mathbf{z}_{m,j} = g(f(\mathbf{x}_{m,j}))$. Then, the contrastive loss for the mini-batch of $M$ images is defined as

$$
\begin{aligned}
\mathcal{L}^{\text{SimCLR}}\left(\{(\mathbf{z}_{m,1}, \mathbf{z}_{m,2})\}_{m=1}^M\right) = -\frac{1}{2M} \sum_{m=1}^M &\log \frac{\exp(\text{sim}(\mathbf{z}_{m,1}, \mathbf{z}_{m,2})/\tau)}{\sum_{j,k} \mathbb{1}_{[\mathbf{z}_{k,j} \neq \mathbf{z}_{m,1}]} \exp(\text{sim}(\mathbf{z}_{m,1}, \mathbf{z}_{k,j})/\tau)} \\
&+ \log \frac{\exp(\text{sim}(\mathbf{z}_{m,2}, \mathbf{z}_{m,1})/\tau)}{\sum_{j,k} \mathbb{1}_{[\mathbf{z}_{k,j} \neq \mathbf{z}_{m,2}]} \exp(\text{sim}(\mathbf{z}_{m,2}, \mathbf{z}_{k,j})/\tau)},
\end{aligned}
\tag{1}
$$

where $\text{sim}$ is a measure of similarity (e.g., cosine similarity) and $\mathbb{1}_{[\mathbf{z}_{k,j} \neq \mathbf{z}_{m,1}]} \in \{0, 1\}$ is an indicator function. The temperature $\tau > 0$ is a hyperparameter controling the sharpness of the distribution.

### 3.3 SELF-SUPERVISED SET REPRESENTATION LEARNING WITH SIMCLR

Existing self-supervised learning has focused on instance-level visual features. The importance of the instance-level features is clear for generalization on unseen tasks, however, a meta-learning problem is often interpreted as a *set*-level problem rather than instance-level. For example, Snell et al. (2017)

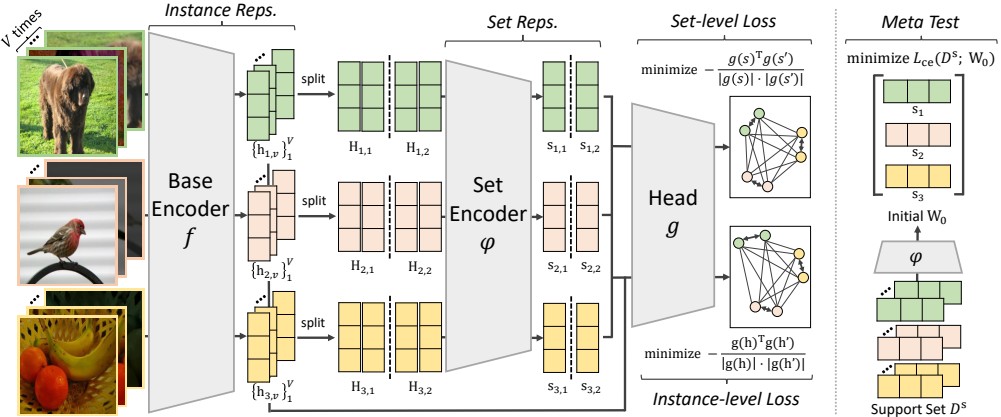

Figure 1: **(Left)**: A conceptual illustration of Set-SimCLR with three images. We first encode each augmented image into instance representation using the base encoder $f$. Then we partition the set of $V$ augmented images into two sets and obtain set representations with the set encoder $\varphi$. We finally compute set- and instance-level loss. We additionally minimize the cross loss in Eq. 3, which is abbreviated in this figure. **(Right)**: At meta-test, we use set representation of each class as an initialization of linear classifier weight.

takes the average of features belonging to the same class as a prototype (or a set representation), or Gordon et al. (2019); Iakovleva et al. (2020) propose Bayesian framework to learn stochastic prototypes using multi-layer perceptron. Further, Triantafillou et al. (2019) propose to fine-tune a prototype with a supervised loss. Inspired by the successes, we propose a self-supervised contrastive learning framework for learning *set representation* to more effectively address UML problems.

**Set Representation**    The underlying assumption of self-supervised learning methods is that two different views of an image share most of the visual semantics. We extend this idea to set-level representation by constructing two sets where each set consists of multiple different views of the same image and maximizing agreement between the two sets. Specifically, we repeatedly apply stochastic augmentations to an image for $V$ times and construct a set $\{\mathbf{x}_{m,v}\}_{v=1}^{V}$ for each image of a mini-batch, where $V$ is an even number. Then we independently encode each augmented image with the base encoder $f$ to obtain instance-level feature representations $\mathbf{h}_{m,v} = f(\mathbf{x}_{m,v}) \in \mathbb{R}^d$ for $m = 1, \ldots, M$ and $v = 1, \ldots, V$, where $d$ is the dimension of the feature representation. This results in $M$ different sets of instance-level representations $\mathbf{H}_m = \{\mathbf{h}_{m,v}\}_{v=1}^{V}$ for $m = 1, \ldots, M$. Then we divide each set $\mathbf{H}_m$ by half to obtain positive pairs of sets, i.e., $\mathbf{H}_{m,1} = \{\mathbf{h}_{m,v}\}_{v=1}^{V/2}$ and $\mathbf{H}_{m,2} = \{\mathbf{h}_{m,v}\}_{v=V/2+1}^{V}$, and get a set representation by applying a set encoder to each set. Any permutation-invariant set encoder that takes a set of vectors as an input and outputs a vector can be employed. Here, we design a set encoder with self-attention for better representation:

$$\mathbf{T}_{m,j} = \mathtt{TransformerEncoder}(\mathbf{H}_{m,j}) \in \mathbb{R}^{V/2 \times d}$$

$$\mathbf{s}_{m,j} = \mathtt{MLP}(\mathtt{concat}\,(\mathtt{mean}(\mathbf{T}_{m,j}); \mathtt{std}(\mathbf{T}_{m,j}); \mathtt{max}(\mathbf{T}_{m,j}); \mathtt{min}(\mathbf{T}_{m,j}))) \in \mathbb{R}^d, \tag{2}$$

and we define a set encoding function $\varphi : \mathbf{H}_{m,j} \in \mathbb{R}^{V/2 \times d} \mapsto \mathbf{s}_{m,j} \in \mathbb{R}^d$. For `TransformerEncoder`, we use the multi-head self-attention mechanism proposed by Vaswani et al. (2017); Lee et al. (2019). Please see Appendix A for more detail. We take the row-wise operations on the outputs $\mathbf{T}_{m,j} \in \mathbb{R}^{V/2 \times d}$ of `TransformerEncoder` to compute mean, standard deviation, maximum and minimum (denoted as `mean`, `std`, `max` and `min`), where each results in a $d$-dimensional vector. Then we concatenate them, denoted as `concat`, which is a $4d$-dimensional vector, and feed it into multi-layer perceptron `MLP` to obtain the final set representation $\mathbf{s}_{m,j} \in \mathbb{R}^d$.

**Contrastive Loss for Set Representation Learning**    We now obtain positive pair of set representations $\mathbf{s}_{m,1}$ and $\mathbf{s}_{m,2}$, by applying the set encoder in Eq. 2 to each set $\mathbf{H}_{m,1}$ and $\mathbf{H}_{m,2}$. Following self-supervised literature, we project this set representation into the latent space with the same head $g$ used for instance-level feature learning. Finally, we compute set-level contrastive loss by plugging the projected set representations into Eq. 1, i.e., $\mathcal{L}^{\text{SimCLR}}(\{(g(\mathbf{s}_{m,1}), g(\mathbf{s}_{m,2})\}_{m=1}^{M})$. The difference between our loss and SimCLR loss is that instead of instance-level representation, we pull positive pair of set-level representation and repulse the negative set pairs. Further, we introduce a cross loss that regularizes the subspace of instance- and set-level representations to be shared in the latent space

as follows: $\mathcal{L}^{\text{SimCLR}}(\{(g(\mathbf{s}_{m,1}), g(\mathbf{h}_{m,2})\}_{m=1}^M)$. Then the final loss is a combination of the set-level and the instance-level SimCLR losses as follows:

$$\underbrace{\mathcal{L}^{\text{SimCLR}}\left(\{(g(\mathbf{h}_{m,1}), g(\mathbf{h}_{m,2}))\}_{m=1}^M\right)}_{\text{Instance-level Loss}} + \underbrace{\mathcal{L}^{\text{SimCLR}}\left(\{(g(\mathbf{s}_{m,1}), g(\mathbf{s}_{m,2}))\}_{m=1}^M\right)}_{\text{Set-level Loss}}$$
$$+ \underbrace{\mathcal{L}^{\text{SimCLR}}\left(\{(g(\mathbf{s}_{m,1}), g(\mathbf{h}_{m,2}))\}_{m=1}^M\right)}_{\text{Cross Loss}} \tag{3}$$

**Linear Evaluation for Downstream Tasks** We now describe how we utilize the learned instance-level and set-level representations on downstream few-shot classification tasks. For a $N$-way $S$-shot task at meta-test time, we are given the support set $\mathcal{D}^s = \{(\mathbf{x}_i^s, y_i^s)\}_{i=1}^{N \times S}$ and supposed to predict the labels of the query set $\mathcal{D}^q = \{\mathbf{x}_i^q\}_{i=1}^{N \times Q}$. We first apply the base encoder $f$ to obtain instance feature representations of the support set $\{(\mathbf{h}_i^s, y_i^s)\}_{i=1}^{N \times S}$ and the query set $\{\mathbf{h}_i^q\}_{i=1}^{N \times Q}$. For each class $c = 1, \ldots, N$, we encode $\mathbf{H}_c^s = \{\mathbf{h}_i^s \mid y_i^s = c\}$, a set of instances belonging to the class $c$, with the mapping $\varphi$ as described in Eq. 2. Let the set representation of $c$-th class be $\mathbf{s}_c = \varphi(\mathbf{H}_c^s) \in \mathbb{R}^d$.

We then initialize a weight of a linear classifier with the set representations $\mathbf{s}_1, \ldots, \mathbf{s}_N$ and train the classifier with the support set while freezing the base encoder $f$, which is similar to the linear evaluation of self-supervised learning (Chen et al., 2020a). We find that this is more suitable for our few-shot setting than the strategy of finetuning the full model to prevent the risk of overfitting to few data. Specifically, we initialize the weight $W$ of the classifier by stacking the learned set representation $s_c$ as row vectors, denoted as $W_0$, and optimize it by minimizing the cross-entropy loss with weight-decay as follows:

$$\underset{W}{\text{minimize}} \, \mathcal{L}_{\text{CE}}(W; \mathcal{D}^s) \text{ via algorithm } \mathcal{A} \text{ as } W^* = \mathcal{A}(\mathcal{L}_{\text{CE}}; W_0, \mathcal{D}^s)$$

$$\mathcal{L}_{\text{CE}}(W; \mathcal{D}^s) = \frac{1}{|\mathcal{D}^s|} \sum_{\mathbf{x}_i^s, y_i^s \in \mathcal{D}^s} \ell\left(Wf(\mathbf{x}_i^s), y_i^s\right) \tag{4}$$

where $W_0 = [\mathbf{s}_1 \cdots \mathbf{s}_N]^\top \in \mathbb{R}^{N \times d}, \ell(q, y) = -\log\left(\exp(q_y) / \sum_{k=1}^N \exp(q_k)\right)$ and $\mathcal{A}(\mathcal{L}_{\text{CE}}; W_0, \mathcal{D}^s)$ denotes an iterative optimization algorithm with weight-decay and the initialization $W_0$. After the optimization, we predict a label for each instance in the query set $\mathcal{D}^q$ as $y_i^q = \arg\max_c p_c^{(i)}$, where $(p_1^{(i)}, \ldots, p_N^{(i)})^\top = W^* f(\mathbf{x}_i^q)$. We provide pseudo-code of our meta-training (self-supervised learning) and meta-test in Appendix B.

**Connection to Meta-Learning** We further discuss why our set representation boosts generalization performance in the view of meta-learning. One of the most effective approaches in meta-learning literature to tackle few-shot learning problems is to learn an initialization and adapt the initialization to meta-test tasks. For instance, ANIL (Raghu et al., 2019) learns a feature extractor and an *initialization of a linear classifier* such that the learned linear classifier can rapidly adapt to the target task while freezing the feature extractor. ANIL has shown that meta-learning the initialization of the linear classifier is crucial for improving the generalization performance of meta-test tasks. In this point of view, Set-SimCLR meta-learns set representations based on the set-level contrastive learning loss using pseudo tasks constructed by leveraging data augmentation, where different views of an image belong to the same pseudo-class. Then the meta-learned set representations are utilized as an initialization which leads to better generalization performance of meta-test tasks. We further provide theoretical motivation of how our set representation can improve generalization in the next section, and a detailed relationship to the meta-learning in Appendix D.

### 3.4 THEORETICAL MOTIVATION

In this section, we provide theoretical motivations on our algorithmic design. In appendix C.1, we show that the proposed method is equivalent to the metric-based inference with the fine-tuning of the class prototypes $\mathbf{s}_c$, where the initial class prototypes $\mathbf{s}_c$ are obtained with the set representation by $\mathbf{s}_c = \varphi(\mathbf{H}_c^s)$ and each input $\mathbf{x}$ is represented by instance-level representation $f(\mathbf{x})$. Thus, in the following, we discuss how such metric-based inference behaves with respect to the supervised loss in the downstream task.

To obtain theoretical insights, this section focuses on the binary classification without the head $g$ and considers the following abstract data-generating process: each of the unknown labels

$y^+$ and $y^-$ is drawn independently from a uniform distribution $U$ on $\{1, 2\}$, and then each of the unlabeled positive examples $\mathbf{x}^+$ and $\mathbf{x}^{++}$ is drawn from the conditional distribution $\mathcal{D}_{y^+}$ conditioned on the label $y^+$ while the negative example $\mathbf{x}^-$ is drawn from the conditional distribution $\mathcal{D}_{y^-}$. Accordingly, this hidden process forms the joint distribution $\mathcal{D}(x, y) = \mathcal{D}_y(x)U(y)$. In this setting, we can write the contrastive unsupervised loss $L^{\text{SimCLR}}$ of the representation $f$ and the corresponding supervised loss $L_{\text{s}}$ of our classifier $W_t f$ by $L^{\text{SimCLR}}(f) = \mathbb{E}_{y^+ \sim U, y^- \sim U}\mathbb{E}_{\mathbf{x}^+, \mathbf{x}^{++} \sim \mathcal{D}_{y^+}^2, \mathbf{x}^- \sim \mathcal{D}_{y^-}}[-\log(\frac{\exp(f(\mathbf{x}^{++})^\top f(\mathbf{x}^+))}{\exp(f(\mathbf{x}^{++})^\top f(\mathbf{x}^+)) + \exp(f(\mathbf{x}^{++})^\top f(\mathbf{x}^-))})]$ and $L_{\text{s}}^t(f) = \mathbb{E}_{(\mathbf{x}, y) \sim \mathcal{D}}[\ell(W_t f(\mathbf{x}), y)]$ where $\ell(q, y) = -\log\left(\frac{\exp(q_y)}{\sum_{k=1}^2 \exp(q_k)}\right)$ and the matrix $W_t \in \mathbb{R}^{2 \times d}$ is defined by $W_t = [\varphi[\mathbf{H}_1^s] + (\vec{\Delta}_t)_1, \varphi[\mathbf{H}_2^s] + (\vec{\Delta}_t)_2]^\top$. Here, $\vec{\Delta}_t = [(\vec{\Delta}_t)_1, (\vec{\Delta}_t)_2]^\top = W_t - W_0 \in \mathbb{R}^{2 \times d}$ is the elements added during the training with the support set. Importantly, $y^+$ and $y^-$ can be the same as $y^+ = y^-$ since we do not know the true labels in the unsupervised loss. This is reflected by the fact that they are sampled from the same (unknown) probability measure on labels $U$. We define the training loss $\hat{L}_{\text{s}}^t(f) = \frac{1}{|\mathcal{D}^s|}\sum_{(\mathbf{x}_i^s, y_i^s) \in \mathcal{D}^s}\ell(W_t f(\mathbf{x}_i^s), y_i^s)$ and the corresponding training loss with the average pooling (instead of our set representation) by $\hat{L}_{\text{s}}^A(f) = \frac{1}{|\mathcal{D}^s|}\sum_{(\mathbf{x}_i^s, y_i^s) \in \mathcal{D}^s}\ell(A f(\mathbf{x}_i^s), y_i^s)$ where $A = [\mathbb{E}_{\mathbf{x} \sim \mathcal{D}_1}[f(\mathbf{x})], \mathbb{E}_{\mathbf{x} \sim \mathcal{D}_2}[f(\mathbf{x})]]^\top \in \mathbb{R}^{2 \times d}$. Define the probability of $y^+$ and $y^-$ being the same by $\mathbb{P}(y^+ = y^-) = \mathbb{E}_{y^+, y^- \sim U^2}[\mathbb{1}\{y^+ = y^-\}]$. Similarly, $\mathbb{P}(y^+ \neq y^-) = \mathbb{E}_{y^+, y^- \sim U^2}[\mathbb{1}\{y^+ \neq y^-\}]$. We define $c = \mathbb{P}(y^+ \neq y^-)^{-1}$ and $\zeta = c \cdot \mathbb{P}(y^+ = y^-)\log(2)$. Let $L_\ell$ be the Lipschitz constant of $\ell$ w.r.t. its first argument. Let $C_\ell$ be upper bounds on $\ell$. Define $C_f = \mathbb{E}_{\mathbf{x}}[\|f(\mathbf{x})\|_2^2]$. The following theorem provides an upper bound on the expected supervised loss $L_{\text{s}}^t(f)$ in the downstream task:

**Theorem 1.** *Let $\Delta_t \in \mathbb{R}_{\geq 0}$ and suppose that $W_t$ satisfies $\|W_t - W_0\|_F \leq \Delta_t$. Then, for any $\delta > 0$, with probability at least $1 - \delta$,*

$$L_{\text{s}}^t(f) \leq cL^{\text{SimCLR}}(f) - \zeta\log(2) - \hat{\gamma}_t + \Delta_t\sqrt{\frac{16L_\ell^2 C_f}{|\mathcal{D}^s|}} + 2C_\ell\sqrt{\frac{\ln(2/\delta)}{2|\mathcal{D}^s|}}. \tag{5}$$

*where $\hat{\gamma}_t = \hat{L}_{\text{s}}^A(f) - \hat{L}_{\text{s}}^t(f)$ and $\|\cdot\|_F$ denotes Frobenius norm.*

The proof is deferred to appendix C.2. Theorem 1 shows that we can minimize the expected supervised loss $L_{\text{s}}^t(f)$ by minimizing the contrastive loss $L^{\text{SimCLR}}(f)$ and training loss $\hat{L}_{\text{s}}^t(f)$. As we increase $t \in \mathbb{N}_0$, the value of $\hat{\gamma}_t = \hat{L}_{\text{s}}^A(f) - \hat{L}_{\text{s}}^t(f)$ increases since $\hat{L}_{\text{s}}^t(f)$ decreases in $t$ while $\hat{L}_{\text{s}}^A(f)$ is a constant in $t$. However, increasing $t$ can also increase $\Delta_t$ in Theorem 1. Thus, there is a tradeoff of $\hat{\gamma}_t$ v.s. $\Delta_t$. At $t = 0$, we have $\Delta_t = 0$. As we increase $t$, both $\hat{\gamma}_t$ and $\Delta_t$ tend to increase. Here, if $|\mathcal{D}^s|$ is very large, then an optimal strategy would be to increase $t$ towards infinity, because the term of $\Delta_t$ is $O(\Delta_t/\sqrt{|\mathcal{D}^s|})$. However, when $|\mathcal{D}^s|$ is small, we do not want to increase $\Delta_t$ too much. Thus, Theorem 1 predicts that we should conduct fine-tuning to control the tradeoff between $\hat{\gamma}_t$ and $\Delta_t$ with the initialization obtained through the unsupervised meta-learning step. We can see in the definition of $\Delta_t$ that the initialization matters to avoid increasing $\Delta_t$ too much while increasing $\hat{\gamma}_t$.

## 4 EXPERIMENT

In this section, we empirically validate the effectiveness of our set representation learning framework on several downstream few-shot classification tasks, and compare our Set-SimCLR against UML baselines and instance-level self-supervised baselines in subsection 4.1 and 4.2, respectively.

### 4.1 COMPARISON TO UNSUPERVISED META-LEARNING

**Dataset** We use the Mini-ImageNet dataset introduced by Ravi & Larochelle (2017), which is a subset of ILSVRC-2012 (Deng et al., 2009). It consists of 100 classes and each class contains 600 different images. We use the resolution of $3 \times 84 \times 84$, which is widely used in the meta-learning literature. We use 64 classes for unsupervised meta-training, 16 classes for meta-validation, and the remaining 20 classes for meta-test. Following the standard protocol of unsupervised meta-learning, we evaluate our method on 1000 randomly sampled tasks from the meta-test set.

**Baselines** We compare Set-SimCLR with four UML methods as the baselines: **1) CACTUs** (Hsu et al., 2019), **2) UMTRA** (Khodadadeh et al., 2019), **3) LASIUM** (Khodadadeh et al., 2021) and **4) Meta-GMVAE** (Lee et al., 2021). In addition, we provide the performance of two supervised meta-learning methods as "*oracles*": **MAML (oracle)** (Finn et al., 2017) and **ProtoNets (oracle)** (Snell et al., 2017). The detailed explanation of the baselines is in Appendix F.

Table 1: Results for 5-way $S$-shot classification on Mini-ImageNet. The base encoder is either Conv4 or Conv5. We report mean and standard deviation of accuracy evaluated on 1000 episodes with 5 different runs for ours. Note that we take the accuracy of baselines from the previous works Khodadadeh et al. (2021); Lee et al. (2021).

| Method | Clustering | 1-shot | 5-shot | 20-shot | 50-shot |
|---|---|---|---|---|---|
| *Training from Scratch* | N/A | 27.59 | 38.48 | 51.53 | 59.63 |
| CACTUs-MAML | BiGAN | 36.24 | 51.28 | 61.33 | 66.91 |
| CACTUs-ProtoNets | BiGAN | 36.62 | 50.16 | 59.56 | 63.27 |
| CACTUs-MAML | ACAI/DC | 39.90 | 53.97 | 63.84 | 69.64 |
| CACTUs-ProtoNets | ACAI/DC | 39.18 | 53.36 | 61.54 | 63.55 |
| UMTRA | N/A | 39.93 | 50.73 | 61.11 | 67.15 |
| LASIUM-MAML-RO/N | N/A | 40.19 | 54.56 | 65.17 | 69.13 |
| LASIUM-ProtoNets-RO/N | N/A | 40.05 | 52.53 | 59.45 | 61.43 |
| Meta-GMVAE | N/A | 42.82 | 55.73 | 63.14 | 68.26 |
| **Set-SimCLR (ours)** | N/A | **43.36** $\pm.34$ | **58.68** $\pm.43$ | **69.12** $\pm.17$ | **73.91** $\pm.36$ |
| *MAML (oracle)* | N/A | 46.81 | 62.13 | 71.03 | 75.54 |
| *ProtoNets (oracle)* | N/A | 46.56 | 62.29 | 70.05 | 72.04 |

Figure 2: **(a)**: 5-way 5-shot classification results on six datasets. **(b)**: 5-way 1, 5, 20, 25-shot classification results on Mini-ImageNet dataset. The base encoder is ResNet-18. We report mean and standard deviation of accuracy evaluated on 1000 episodes with 5 different runs. See Appendix L for the results in tabular format.

**Implementation Details**    We use Conv5 architecture as the base encoder for the fair comparison. We provide the details of neural architectures for base encoder, set encoder and head in Appendix H. We follow SimCLR (Chen et al., 2020a;b) for random augmentation, which is detailed in Appendix J. We apply the composed augmentations to 64 mini-batch images eight times (*i.e.*, $M = 64, V = 8$), resulting in 4 elements in each set. We optimize the base encoder, set encoder and head network for 400 epochs using Adam optimizer (Kingma & Ba, 2015) with default settings (*i.e.*, $\beta_1 = 0.9$ and $\beta_2 = 0.999$). We use constant learning rate of 0.001. For downstream tasks, we use L-BFGS (Liu & Nocedal, 1989) algorithm implemented in scikit-learn (Pedregosa et al., 2011) package to optimize a linear classifier.

**Results**    Table 1 shows the performance of the baselines and our Set-SimCLR for 5-way 1, 5, 20 and 50-shot classification on the Mini-ImageNet dataset, where Set-SimCLR outperforms all the baselines by considerable margins. For an instance, it achieves $+0.54\%, +2.95\%, +3.95\%$, and $+4.27\%$ performance improvement over the best performing baseline on 1- 5-, 20- and 50-shot settings. Notably, the performance gain of Set-SimCLR over the baselines gets larger as we increase the number of instances for a support set, *i.e.*, shot. We can observe the similar pattern when comparing MAML-variant and ProtoNet-variant within baselines, *e.g.*, CACTUs-MAML vs CACTUs-ProtoNets and LASIUM-MAML-RO/N vs LASIUM-ProtoNets-RO/N. This is because the adaptation with the support set at meta-test gets more effective since the model is less likely to overfit to larger shot.

## 4.2 COMPARISON TO SELF-SUPERVISED LEARNING (SSL)

**Dataset**    We use the Mini-ImageNet dataset for training and evaluating models. Further, to verify the effectiveness of the proposed method on transfer learning scenarios, we evaluate the models trained with Mini-ImageNet on the conventional meta-test split of Tiny-ImageNet (Le & Yang, 2015), CIFAR100 (Krizhevsky et al., 2009), Aircraft (Maji et al., 2013), Stanford Cars (Krause et al., 2013) and CUB (Wah et al., 2011) datasets. See Appendix E for the number of classes of meta-splits for each dataset. Since all the models are trained on $84 \times 84$ images from the source dataset Mini-ImageNet, we resize the image to $84 \times 84$ resolution for all the target datasets. Following UML literature, we evaluate our method on 1,000 randomly sampled tasks from the meta-test set.

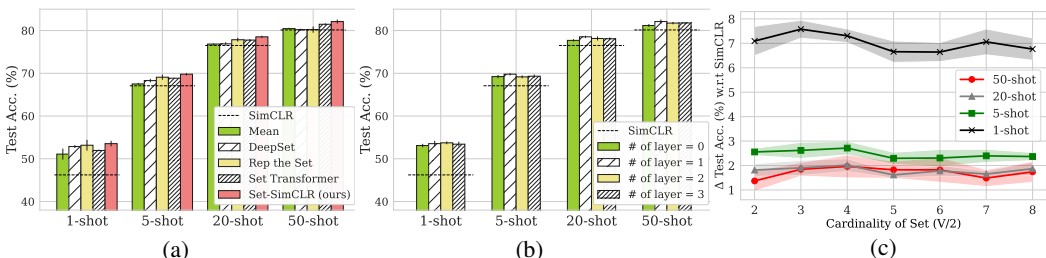

Figure 3: The results of ablation study on the 5-way 1, 5, 20, 50-shot classification using Mini-ImageNet. We study the effectiveness of **(a)**: different set encoder architectures, **(b)**: the depth of `TransformerEncoder` layers, and **(c)**: the number of set elements w.r.t SimCLR. We report the results over 3 different random seeds.

**Baselines**   Although there are a vast number of SSL methods, in this work, we want to show the effectiveness of SSL compared to learning instance representation. Thus, we choose following four representative contrastive SSL baselines as follows: **1) SimCLR (Chen et al., 2020a;b)**, **2) MOCO (He et al., 2020; Chen et al., 2020c; 2021)**, **3) BYOL (Grill et al., 2020)**, **4) Barlow Twins (Zbontar et al., 2021)**.  All the details are deferred to Appendix F. Note that we have tried a very recent SSL method — MAE (He et al., 2022), however, it fails to achieve comparable performance to ours and baselines. Please see details in Appendix G.

**Implementation Details**   For the base encoder $f$, we use ResNet-18 architecture (He et al., 2016) which is widely used for evaluating self-supervised learning methods. For a fair comparison, we use the same architecture of head network $g$, for all SSL methods except for MOCO since MOCO does not use the head. For our method Set-SimCLR, we apply the augmentations (which is defined in Appendix J) 8 times to the mini-batch of 64 images (*i.e.*, $M = 64, V = 8$), resulting in 4 elements in each set, while performing the same augmentation twice on the mini-batch of 256 images (*i.e.*, $M = 256, V = 2$) for the baselines. Following SSL literature, we train a linear classifier for downstream tasks using scikit-learn package with default settings. We provide more implementation details in Appendix I.

**Results**   Figure 2a shows 5-way 5-shot experimental results of all the models on the Mini-ImageNet, Tiny-ImageNet, CIFAR100, Aircraft, Stanford Cars, CUB datasets. We can see that Set-SimCLR outperforms all the SSL baselines by considerable margins, from $+0.17\%$ to $+2.71\%$. The results of Set-SimCLR in the transfer learning scenario, one of the important goals of the self-supervised learning methods, is particularly remarkable. We further evaluate ours and baselines over various shots, *e.g.*, 1-, 10-, 20 and 50-shot on the Mini-ImageNet. As shown in Figure 2b, our Set-SimCLR obtains outstanding performance gains of $7.31\%, 2.71\%, 2.02\%, 1.96\%$ over the best performing baselines on 1-, 10-, 20-, 50-shot settings. Notably, that performance gain of Set-SimCLR is much larger in 1-shot setting than the other shots. It shows that SSL baselines are vulnerable to overfitting to the single shot. In contrast, the classifiers obtained by Set-SimCLR shows much robustness in the 1-shot setting due to the initialization with learned set representations.

### 4.3 ABLATION STUDY AND ANALYSIS

In this subsection, we conduct ablation studies to verify a necessity of each components. We further provide analysis on our Set-SimCLR in comparison to SSL baselines.

**Set Encoder Architecture**   We replace the architecture of the set encoder described in Eq. 2 with mean pooling, Deep Set (Zaheer et al., 2017), Rep the Set (Skianis et al., 2020), or Set Transformer (Lee et al., 2019). Figure 3a shows the 5-way 5-shot test accuracy of different set encoder architectures on the Mini-ImageNet dataset. We find that Rep the Set architecture works well on 1-shot setting, and our set encoder $\varphi$ in Equation 2 shows slightly better performance on 5-, 20- and 50-shot settings than the others. Note that our Set-SimCLR is set representation learning framework that is agnostic to the choice of set encoder architecture. Furthermore, even with the simplest architecture (mean pooling), it still shows slightly better performance than the best-performing self-supervised baseline (SimCLR) which is denoted as dotted lines.

**The Depth of Set Encoder**   Another important compoment of our model is the number of `TransformerEncoder` layers in Equation 2. First, we start without `TransformerEncoder` layer, *i.e.*, identity function, and increase the depth of the set encoder. Figure 3b shows the 5-way 5-shot test accuracy on the Mini-ImageNet dataset with varying the number of layers. We find that the set

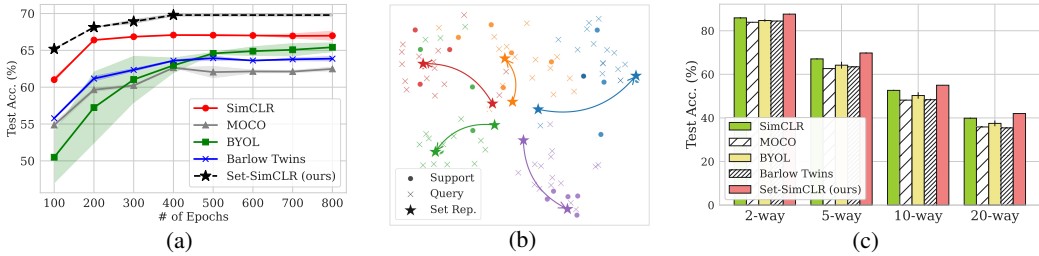

Figure 4: Analysis of the proposed Set-SimCLR on the Mini-ImageNet dataset. **(a)**: 5-way 5-shot test accuracy of baselines as a function of training epochs. **(b)**: T-SNE visualization of our adaptation process on a 5-way 5-shot task. **(c)**: 5-shot test accuracy of different ways. We report the results over 3 different runs.

encoder with a single layer is the most effective on the overall settings considering the computational cost due to extra layers. Note that all of our models with the different number of layers outperform the best performing self-supervised baseline (SimCLR) which is denoted as dotted lines.

**Cardinality of Set**    In order to study effects of the number of set elements for Set-SimCLR, we plot 5-way 5-shot test accuracy improvement over SimCLR, denoted as $\Delta$ Test Acc., as a function of the cardinality of set. In Figure 3c, the performance of the downstream tasks is not sensitive to the size of sets, which results in consistent improvement over SimCLR with all the cardinality we consider.

**Training Budgets Analysis**    It approximately takes twice longer to train our Set-SimCLR than the baselines, since it requires multiple stochastic augmentations to construct a set (Please see wall-clock time in Appendix K). Then one may wonder whether the baseline can be comparable or even better if we train it with similar computational budgets to ours. To address this question, we train the self-supervised learning baselines for 800 epochs, which is twice larger than the before, and observe test accuracy over training. Figure 4a shows the 5-way 5-shot test accuracy of self-supervised learning baselines on the Mini-ImageNet dataset. We find that the our Set-SimCLR evaluated at 400 epochs largely outperforms the self-supervised baselines for all the training budgets we consider.

**Qualitative Analysis on Adaptation of Set Representation**    We now qualitatively analyze the adaptation process of our set representation at meta-test time. To do so, we visualize the set representations before and after the adaptation (*i.e.*, each row of the classifier weight $W_0$ and $W^*$), and instances from support and query set. We normalize all the examples to be length 1 and project them to 2d space with T-SNE (Van der Maaten & Hinton, 2008). Figure 4b shows instance representation from query and support set and set representations, denoted as circle, cross and star, respectively. We represent arrows as the adaptation process of set representation, and the color stands for each class. We find that the set representation is not that discriminative at the beginning, however, it represents each class very well after the adaptation. This shows the necessity of our proposed adaptation process of set representation to achieve better performance of the downstream tasks.

**Accuracy with Various Ways**    We finally conduct experiments to show the performance of each model with varying the way of meta-test tasks. Figure 4c shows the 2-, 5-, 10- and 20-way 5-shot test accuracy of the self-supervised learning baselines and ours on the Mini-ImageNet dataset. We find that our Set-SimCLR consistently outperforms the baselines on all the way we consider here.

## 5    CONCLUSION

In this paper, we proposed self-supervised set representation learning framework for unsupervised meta-learning (UML). Our Set-SimCLR learns set representation by maximizing the agreement between positive sets in latent space, where the positive sets are constructed with repeated stochastic augmentations of an image. Based on theoretical analysis, we studied how the learned set representation can improve generalization ability and why it makes sense to initialize of the weight of linear classifier with the learned set representation for downstream tasks. We further validated the empirical efficacy of proposed Set-SimCLR and compared it against UML and self-supervised baselines using several benchmark few-shot classification datasets. Note that our main idea of minimizing distance between semantically similar sets constructed with repeated augmentations is not limited to SimCLR framework. Based on this, we plan to expand our framework to various self-supervised learning methods to exploit their potential merits.

## REPRODUCIBILITY STATEMENT

We clearly specify implementation details for reproducibility, including data split, baselines for comparisons, neural architecture, training process and augmentation in Appendix E, F, H, I, and J. In Supplementary File, we further provide the code for reproducing the main experimental results in Table 1 and Figure 2. Note that all the numerical results are based on more than three runs. Lastly, we will release our full code and the checkpoint of models to be publicly available after acceptance.

## ACKNOWLEDGEMENTS

This work was supported by Institute of Information & communications Technology Planning & Evaluation (IITP) grant funded by the Korea government(MSIT) (No.2019-0-00075, Artificial Intelligence Graduate School Program(KAIST)), Institute of Information & communications Technology Planning & Evaluation (IITP) grant funded by the Korea government(MSIT) (No.2022-0-00713), KAIST-NAVER Hypercreative AI Center, the Engineering Research Center Program through the National Research Foundation of Korea (NRF) funded by the Korean Government MSIT (NRF-2018R1A5A1059921), and Samsung Electronics (IO201214-08145-01).

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

## A    TRANSFORMER ENCODER

We describe the TransformerEncoder from Eq. 2 in more detail. Let $X \in \mathbb{R}^{n \times d}$ be a stack of $n$ $d$-dimensional row vectors. Let $W_j^Q, W_j^K, W_j^V \in \mathbb{R}^{d \times d_k}$ be weight matrices for self-attention and let $b_j^Q, b_j^K, b_j^V \in \mathbb{R}^{d_k}$ be bias vectors for $j = 1, \ldots, 4$. For encoding the input $X$, we compute self-attention as follows:

$$Q^{(j)} = XW_j^Q + \mathbf{1}(b_{1,j}^Q)^\top \in \mathbb{R}^{n \times d_k}$$
$$K^{(j)} = XW_j^Q + \mathbf{1}(b_{1,j}^K)^\top \in \mathbb{R}^{n \times d_k}$$
$$V^{(j)} = XW_j^K + \mathbf{1}(b_{i,j}^V)^\top \in \mathbb{R}^{n \times d_k}$$
$$A^{(j)}(X) = \text{LayerNorm}\left( Q_1^{(j)} + \text{softmax}\left( Q_1^{(j)}(K_1^{(j)})^\top / \sqrt{d_k} \right) V_1^{(j)} \right) \in \mathbb{R}^{n \times d_k} \tag{6}$$
$$O(X) = \text{Concat}(A_1^{(1)}(X), \ldots, A_1^{(4)}(X)) \in \mathbb{R}^{n \times d}$$

where $\mathbf{1} = (1, \ldots, 1)^\top \in \mathbb{R}^n$ is a vector of ones, $d = 4d_k$, and softmax is applied for each row. After self-attention, we add a skip connection with layer normalization (Ba et al., 2016) as follows:

$$\texttt{TransformerEncoder}(X) = \text{LayerNorm}\left(O(X)\right) + \text{ReLU}\left(WO(X) + \mathbf{1}b^\top\right) \tag{7}$$

where $W \in \mathbb{R}^{d \times d}, b \in \mathbb{R}^d$.

## B    ALGORITHM

We provide the pseudo-code for Set-SimCLR described in Section 3.3.

---

**Algorithm 1** Meta-Training for Set-SimCLR

1: **Input:** Batch size $M$, constant $\tau$, the number of augmentations $V$, augmentation $\mathcal{T}$, and unlabeled dataset $\mathcal{D}^u$
2: **while** not converged **do**
3:    Sample a mini batch $\{\mathbf{x}_m\}_{m=1}^M$ from $\mathcal{D}^u$
4:    **for** $m \leftarrow 1, \ldots, M$ **do**
5:       **for** $v \leftarrow 1, \ldots, V$ **do**
6:          Sample augmentation functions $t \sim \mathcal{T}$
7:          $\mathbf{x}_{m,v} \leftarrow t(\mathbf{x}_m)$
8:          $\mathbf{h}_{m,v} \leftarrow f(\mathbf{x}_{m,v})$
9:       **end for**
10:       $\mathbf{H}_{m,1} \leftarrow \{\mathbf{h}_{m,v}\}_{v=1}^{V/2}$
11:       $\mathbf{H}_{m,2} \leftarrow \{\mathbf{h}_{m,v}\}_{v=V/2+1}^{V}$
12:       $\mathbf{s}_{m,1} \leftarrow \varphi(\mathbf{H}_{m,1}), \mathbf{s}_{m,2} \leftarrow \varphi(\mathbf{H}_{m,2})$
13:    **end for**
14:    $\mathcal{L} \leftarrow \mathcal{L}^{\text{SimCLR}}\left(\{(g(\mathbf{h}_{m,1}), g(\mathbf{h}_{m,2}))\}_{m=1}^M\right)$
15:    $\mathcal{L} += \mathcal{L}^{\text{SimCLR}}\left(\{(g(\mathbf{s}_{m,1}), g(\mathbf{s}_{m,2}))\}_{m=1}^M\right)$
16:    $\mathcal{L} += \mathcal{L}^{\text{SimCLR}}\left(\{(g(\mathbf{s}_{m,1}), g(\mathbf{h}_{m,2}))\}_{m=1}^M\right)$
17:    Perform gradient descent on the loss $\mathcal{L}$ w.r.t the parameters of $f, g$, and $\varphi$.
18: **end while**
19: **Output:** $f, \varphi$

**Algorithm 2** Meta-Test for Set-SimCLR

1: **Input:** Support set $\mathcal{D}^s = \{(\mathbf{x}_i^s, y_i^s)\}_{i=1}^{N \times S}$, query set $\mathcal{D}^q = \{\mathbf{x}_i^q\}_{i=1}^{N \times Q}$, a pretrained encoder $f$, and a pretrained set encoder $\varphi$.
2: $W \leftarrow \mathbf{0} \in \mathbb{R}^{d \times N}$
3: $U \leftarrow \{\mathbf{h}_i^s = f(\mathbf{x}_i^s) \in \mathbb{R}^d\}_{i=1}^{N \times S}$
4: $B \leftarrow |\mathcal{D}^s|$
5: **while** not converged **do**
6:    **for** $c \leftarrow 1, \ldots, N$ **do**
7:       $\mathbf{H}_c^s \leftarrow \{\mathbf{h}_i^s \in U \mid y_i^s = c\}$
8:       $\mathbf{s}_c \leftarrow \varphi(\mathbf{H}_c^s)$
9:       $W[c, :] \leftarrow \mathbf{s}_c$
10:    **end for**
11:    $\mathcal{L} \leftarrow \frac{1}{B} \sum_{i=1}^B -\log \frac{\exp((W\mathbf{h}_i^s)_{y_i^s})}{\sum_{k=1}^N \exp((W\mathbf{h}_i^s)_k)}$
12:    Update $W$ to minimize $\mathcal{L}$ with L-BFGS (Liu & Nocedal, 1989).
13: **end while**
14: **for** $i \leftarrow 1, \ldots, N \times Q$ **do**
15:    $y_i^q \leftarrow \arg\max_c W f(\mathbf{x}_i^q)_c$
16: **end for**
17: **Output:** $\{y_i^q\}_{i=1}^{N \times Q}$
18:
19:
20:

---

## C  ON THEORETICAL MOTIVATION

### C.1  ON THE RELATIONSHIP WITH METRIC-BASED INFERENCE

The metric-based inference using the instance-level representation $\mathbf{h}$ of $\mathbf{x}$ with the class prototypes $\mathbf{s}_c$ can be written by

$$\hat{y}(\mathbf{x}) = \arg\min_c d(\mathbf{s}_c, \mathbf{h}).$$

By choosing the metric to be the negative dot product as $d(s_c, h) = -s_c^\top h$, we can write

$$\hat{y}(x) = \arg\min_c -\mathbf{s}_c^\top h = \arg\min_c -\log \frac{\exp(\mathbf{s}_c^\top h)}{\sum_k \exp(\mathbf{s}_k^\top h)} = \arg\min_c \ell(\texttt{softmax}(W_0 f(\mathbf{x})), c),$$

where the second line follows from the fact that the output of $\arg\min_c$ does not change by adding the constant in $c$. In other words, the prediction of the metric-based inference with $d(\mathbf{s}_c, \mathbf{h}) = -\mathbf{s}_c^\top h$ (without further fine-tuning) is equivalent to the proposed method at the initialization of $W_0$. Thus, the proposed method can be understood as the metric-based inference with the fine-tuning of the class prototypes $s_c$ based on the support set, where the initial class prototypes $s_c$ are obtained by the set representation and each input $\mathbf{x}$ represented by instance-level representation $\mathbf{h}$.

In this view, a naive approach of fine-tuning of the class prototypes $\mathbf{s}_c$ is to fine-tune the parameters of $\varphi$ to minimize $-\log \frac{\exp(\mathbf{s}_c^\top \mathbf{h})}{\sum_k \exp(\mathbf{s}_k^\top \mathbf{h})}$ with the support set where $s_c = \varphi(\mathbf{H}_c)$. However, since $\varphi$ has many parameters, changing $\varphi$ allows to change $\mathbf{s}_c = \varphi(\mathbf{H}_c)$ freely without restrictions on the space of $\mathbf{s}_c$. Thus, instead of fine-tuning the parameters of $\varphi$, we can directly optimize the values of $\mathbf{s}_c$ by initializing $\mathbf{s}_c = \varphi(\mathbf{H}_c)$ and untying $\mathbf{s}_c$ from $\varphi(\mathbf{H}_c)$. This is what is done in the proposed algorithm. This results in the faster computation and the well-behaving convex optimization when compared to the fine-tuning of parameters of $\varphi$.

### C.2  PROOF OF THEOREM 1

*Proof.* We define the performance difference between the average pooling and our set representation in terms of the expected loss by $\gamma = L_s^A(f) - L_s^t(f)$. Define the function $\ell$ by

$$\ell(q) = \log(1 + \exp(-q)).$$

Then,

$$\begin{aligned}
&\ell\left(f(\mathbf{x}^{++})^\top(f(\mathbf{x}^+) - f(\mathbf{x}^-))\right) \\
&= \log(1 + \exp(-f(\mathbf{x}^{++})^\top(f(\mathbf{x}^+) - f(\mathbf{x}^-)))) \\
&= \log\left(\left(1 + \exp(-f(\mathbf{x}^{++})^\top f(\mathbf{x}^+) + f(\mathbf{x}^{++})^\top f(\mathbf{x}^-))\right) \times \frac{\exp(f(\mathbf{x}^{++})^\top f(\mathbf{x}^+))}{\exp(f(\mathbf{x}^{++})^\top f(\mathbf{x}^+))}\right) \\
&= \log\left(\frac{\exp(f(\mathbf{x}^{++})^\top f(\mathbf{x}^+)) + \exp(f(\mathbf{x}^{++})^\top f(\mathbf{x}^-))}{\exp(f(\mathbf{x}^{++})^\top f(\mathbf{x}^+))}\right) \\
&= -\log\left(\frac{\exp(f(\mathbf{x}^{++})^\top f(\mathbf{x}^+))}{\exp(f(\mathbf{x}^{++})^\top f(\mathbf{x}^+)) + \exp(f(\mathbf{x}^{++})^\top f(\mathbf{x}^-))}\right).
\end{aligned}$$

Thus, we have that

$$L^{\mathrm{SimCLR}}(f) = \mathbb{E}_{\substack{y^+ \sim U \\ y^- \sim U}} \mathbb{E}_{\substack{\mathbf{x}^+, \mathbf{x}^{++} \sim \mathcal{D}_{y^+}^2 \\ \mathbf{x}^- \sim \mathcal{D}_{y^-}}} \left[\ell\left(f(\mathbf{x}^{++})^\top(f(\mathbf{x}^+) - f(\mathbf{x}^-))\right)\right].$$

Then, from the convexity of $\ell$, Jensen's inequality and the linearity of the expectation, we have that

$$L^{\mathrm{SimCLR}}(f) \geq \mathbb{E}_{\substack{y^+ \sim U \\ y^- \sim U}} \mathbb{E}_{\mathbf{x}^{++} \sim \mathcal{D}_{y^+}} \left[\ell\left(f(\mathbf{x}^{++})^\top(\mathbb{E}_{\mathbf{x}^+ \sim \mathcal{D}_{y^+}}[f(\mathbf{x}^+)] - \mathbb{E}_{\mathbf{x}^- \sim \mathcal{D}_{y^-}}[f(\mathbf{x}^-)])\right)\right].$$

By decomposing the expectation with sums of conditional expectations, conditioned on the event of $y^+ = y^-$ and its complement of $y^+ \neq y^-$,

$$L^{\mathrm{SimCLR}}(f) \geq \mathbb{P}(y^+ = y^-)\kappa_1 + \mathbb{P}(y^+ \neq y^-)\kappa_2 \qquad (8)$$

where

$$\kappa_1 = \mathbb{E}_{\substack{y^+ \sim U \\ y^- \sim U}} \left[ \mathbb{E}_{\mathbf{x}^{++} \sim \mathcal{D}_{y^+}} \left[ \ell \left( f(\mathbf{x}^{++})^\top (\mathbb{E}_{\mathbf{x}^+ \sim \mathcal{D}_{y^+}} [f(\mathbf{x}^+)] - \mathbb{E}_{\mathbf{x}^- \sim \mathcal{D}_{y^-}} [f(\mathbf{x}^-)]) \right) \right] \mid y^+ = y^- \right]$$

and

$$\kappa_2 = \mathbb{E}_{\substack{y^+ \sim U \\ y^- \sim U}} \left[ \mathbb{E}_{\mathbf{x}^{++} \sim \mathcal{D}_{y^+}} \left[ \ell \left( f(\mathbf{x}^{++})^\top (\mathbb{E}_{\mathbf{x}^+ \sim \mathcal{D}_{y^+}} [f(\mathbf{x}^+)] - \mathbb{E}_{\mathbf{x}^- \sim \mathcal{D}_{y^-}} [f(\mathbf{x}^-)]) \right) \right] \mid y^+ \neq y^- \right].$$

For the first term, since $y^+ = y^-$ inside the loss $\ell$, we have that

$$\kappa_1 = \ell(0) = \log(2) \tag{9}$$

For the second term,

$$\kappa_2 = \mathbb{E}_{y^+ \sim U} \left[ \mathbb{E}_{\mathbf{x}^{++} \sim \mathcal{D}_{y^+}} \left[ \ell \left( f(\mathbf{x}^{++})^\top (\mathbb{E}_{\mathbf{x}^+ \sim \mathcal{D}_{y^+}} [f(\mathbf{x}^+)] - \mathbb{E}_{\mathbf{x}^- \sim \mathcal{D}_{\sigma(y+)}} [f(\mathbf{x}^-)]) \right) \right] \right]$$
$$= \mathbb{E}_{(\mathbf{x},y) \sim \mathcal{D}} \left[ \ell \left( g_\varphi(\mathbf{x})_y - g_\varphi(\mathbf{x})_{\sigma(y)} \right) \right] + \tilde{\gamma} \tag{10}$$

where $g_\varphi(\mathbf{x}) = W_t f(\mathbf{x})$, $g_A(\mathbf{x}) = A f(\mathbf{x})$, $\tilde{\gamma} = \mathbb{E}_{(\mathbf{x},y) \sim \mathcal{D}} [\ell \left( g_A(\mathbf{x})_y - g_A(\mathbf{x})_{\sigma(y)} \right) - \ell \left( g_\varphi(\mathbf{x})_y - g_\varphi(\mathbf{x})_{\sigma(y)} \right)]$ and $\sigma$ is defined as

$$\sigma(y) = \begin{cases} 1 & \text{if } y = 2 \\ 2 & \text{if } y = 1, \end{cases}$$

we have that

$$\ell \left( g_\varphi(\mathbf{x})_y - g_\varphi(\mathbf{x})_{\sigma(y)} \right) = \log \left( \left( 1 + \exp(-g_\varphi(\mathbf{x})_y + g_\varphi(\mathbf{x})_{\sigma(y)}) \right) \times \frac{\exp(g_\varphi(\mathbf{x})_y)}{\exp(g_\varphi(\mathbf{x})_y)} \right)$$
$$= -\log \left( \frac{\exp(g_\varphi(\mathbf{x})_y)}{\exp(g_\varphi(\mathbf{x})_y) + \exp(g_\varphi(x)_{\sigma(y)})} \right)$$
$$= -\log \left( \frac{\exp(g_\varphi(\mathbf{x})_y)}{\sum_{k=1}^{2} \exp(g_\varphi(\mathbf{x})_k)} \right). \tag{11}$$

Similarly,

$$\ell(g_A(\mathbf{x})_y - g_A(\mathbf{x})_{\sigma(y)}) = -\log \left( \frac{\exp(g_A(\mathbf{x})_y)}{\sum_{k=1}^{2} \exp(g_A(\mathbf{x})_k)} \right). \tag{12}$$

By combining equation 10, equation 11, and equation 12,

$$\kappa_2 = \mathbb{E}_{(\mathbf{x},y) \sim \mathcal{D}} \left[ -\log \left( \frac{\exp(g_\varphi(\mathbf{x})_y)}{\sum_{k=1}^{2} \exp(g_\varphi(\mathbf{x})_k)} \right) \right] + \gamma, \tag{13}$$

By combining equation 8, equation 9 and equation 13, we have

$$L^{\text{SimCLR}}(f) \geq \mathbb{P}(y^+ \neq y^-) \left( \mathbb{E}_{(\mathbf{x},y) \sim \mathcal{D}} \left[ -\log \left( \frac{\exp(g_\varphi(\mathbf{x})_y)}{\sum_{k=1}^{2} \exp(g_\varphi(\mathbf{x})_k)} \right) \right] + \gamma \right)$$
$$+ \mathbb{P}(y^+ = y^-) \log(2)$$

This implies that

$$L_s^t(f) \leq c L^{\text{SimCLR}}(f) - \zeta \log(2) + (L_s^t(f) - L_s^A(f)).$$

By using Hoeffding's inequality,

$$\mathbb{P} \left( \hat{L}_s^A(f) - L_s^A(f) \geq t \right) \leq \exp \left( -\frac{2t^2}{\sum_{(\mathbf{x}_i^s, y_i^s) \in \mathcal{D}^s} (|\mathcal{D}_s|^{-1} C_\ell)^2} \right) = \exp \left( -\frac{2t^2 |\mathcal{D}^s|}{C_\ell^2} \right)$$

for all $t > 0$. Note that $\mathbb{E} \left[ \hat{L}_s^A(f) \right] = L_s^A(f)$. Let $\delta := \exp(-2t^2 |\mathcal{D}^s| / C_\ell^2)$. Then we get $t = C_\ell \sqrt{\ln(1/\delta)(2|\mathcal{D}^s|)^{-1}}$. In other words, for any $\delta > 0$, with probability at least $1 - \delta$,

$$\hat{L}_s^A(f) - L_s^A(f) \leq C_\ell \sqrt{\frac{\ln(1/\delta)}{2|\mathcal{D}^s|}}.$$

Thus, for any $\delta > 0$, with probability at least $1 - \delta$,

$$L_s^t(f) \leq cL^{\text{SimCLR}}(f) - \zeta \log(2) + (L_s^t(f) - \hat{L}_s^A(f)) + C_\ell \sqrt{\frac{\ln(1/\delta)}{2|\mathcal{D}^s|}}. \tag{14}$$

Let $\mathcal{W}_t = \{W_t \in \mathbb{R}^{2 \times d} : \|W_t - W_0\|_F \leq \Delta_t\}$. Then, since $W_t \in \mathcal{W}_t$ from the assumption on $W_t$, by using Lemma 4 of (Pham et al., 2021), for any $\delta > 0$, with probability at least $1 - \delta$, the following holds:

$$L_s^t(f) \leq \hat{L}_s^t(f) + 2\mathcal{R}_n(\mathcal{W}_t) + C_\ell \sqrt{\frac{\ln(1/\delta)}{2n}}, \tag{15}$$

where $\mathcal{R}_n(\mathcal{W}_t) = \mathbb{E}_{s,\xi}[\sup_{W \in \mathcal{W}_t} \frac{1}{n} \sum_{i=1}^n \xi_i \ell(Wf(\mathbf{x}_i), y_i)]$, $s = ((\mathbf{x}_i, y_i))_{i=1}^n$, $n = |\mathcal{D}^s|$, and $\xi_1, \ldots, \xi_n$ are independent uniform random variables taking values in $\{-1, 1\}$.

Given a matrix $M \in \mathbb{R}^{m \times m'}$, let $\text{vec}[M] \in \mathbb{R}^{mm'}$ be the vectorization of $M$. By using Corollary 4 of (Maurer, 2016),

$$\mathcal{R}_n(\mathcal{W}_t) \leq \frac{\sqrt{2}L_\ell}{n} \mathbb{E}_{s,\xi} \left[ \sup_{W \in \mathcal{W}_t} \sum_{i=1}^n \sum_{k=1}^2 \xi_{ik} W_k f(\mathbf{x}_i) \right]$$

$$= \frac{\sqrt{2}L_\ell}{n} \mathbb{E}_{s,\xi} \left[ \sup_{W \in \mathcal{W}_t} \sum_{k=1}^2 W_k \sum_{i=1}^n \xi_{ik} f(\mathbf{x}_i) \right]$$

$$= \frac{\sqrt{2}L_\ell}{n} \mathbb{E}_{s,\xi} \left[ \sup_{W \in \mathcal{W}_t} w^\top h \right]$$

where $W_k$ is the $k$-th row of $W$, $w = \text{vec}[W^\top] \in \mathbb{R}^{2d}$, $\xi_{ik}$ are independent uniform random variables taking values in $\{-1, 1\}$, $h = \text{vec}[H] \in \mathbb{R}^{2d}$, and $H \in \mathbb{R}^{d \times 2}$ with $H_{jk} = \sum_{i=1}^n \xi_{ik} f(\mathbf{x}_i)_j$. Define $w_0 = \text{vec}[W_0^\top]$. Since $\mathbb{E}_{s,\xi}[w_0^\top h] = w_0^\top \mathbb{E}_{s,\xi}[h] = 0$, we have

$$\mathcal{R}_n(\mathcal{W}_t) \leq \frac{\sqrt{2}L_\ell}{n} \mathbb{E}_{s,\xi} \left[ \sup_{W \in \mathcal{W}_t} w^\top h \right] = \frac{\sqrt{2}L_\ell}{n} \mathbb{E}_{s,\xi} \left[ \sup_{W \in \mathcal{W}_t} w^\top h \right] - \frac{\sqrt{2}L_\ell}{n} \mathbb{E}_{s,\xi} \left[ w_0^\top h \right]$$

$$= \frac{\sqrt{2}L_\ell}{n} \mathbb{E}_{s,\xi} \left[ \sup_{W \in \mathcal{W}_t} (w - w_0)^\top h \right]$$

Thus,

$$\mathcal{R}_n(\mathcal{W}_t) \leq \frac{\sqrt{2}L_\ell}{n} \mathbb{E}_{s,\xi} \left[ \sup_{W \in \mathcal{W}_t} \|w - w_0\|_2 \|h\|_2 \right] = \frac{\sqrt{2}L_\ell \Delta_t}{n} \mathbb{E}_{s,\xi}[\|h\|_2]$$

Here,

$$\mathbb{E}_{s,\xi}[\|h\|_2] = \mathbb{E}_{s,\xi} \sqrt{\sum_{j=1}^d \sum_{k=1}^2 \left( \sum_{i=1}^n \xi_{ik} f(\mathbf{x}_i)_j \right)^2} \leq \sqrt{\sum_{j=1}^d \sum_{k=1}^2 \mathbb{E}_{s,\xi} \left( \sum_{i=1}^n \xi_{ik} f(\mathbf{x}_i)_j \right)^2}$$

$$= \sqrt{\sum_{j=1}^d \sum_{k=1}^2 \mathbb{E}_s \sum_{i=1}^n (f(\mathbf{x}_i)_j)^2}$$

$$= \sqrt{\sum_{k=1}^2 \sum_{i=1}^n \mathbb{E}_s \sum_{j=1}^d (f(\mathbf{x}_i)_j)^2}$$

$$= \sqrt{\sum_{k=1}^2 \sum_{i=1}^n \mathbb{E}_s \|f(\mathbf{x}_i)\|_2^2}$$

$$\leq \sqrt{2C_f n}$$

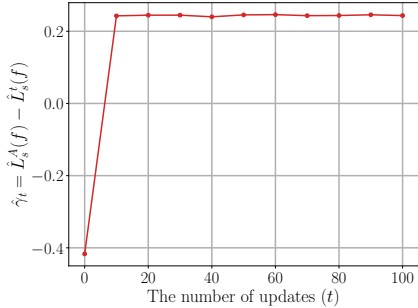

Figure 5: Plotting of the value of $\hat{\gamma}_t = \hat{L}_s^A(f) - \hat{L}_s^t(f)$ as we update the weight $W_t$ with the support set $\mathcal{D}^s$.

Combining these, we have

$$\mathcal{R}_n(\mathcal{W}_t) \leq \frac{L_\ell \sqrt{4C_f}\Delta_t}{\sqrt{n}}. \tag{16}$$

Combining equation 14, equation 15, and equation 16 with union bounds, we have that for any $\delta > 0$, with probability at least $1 - \delta$,

$$L_s^t(f) \leq cL^{\text{SimCLR}}(f) - \zeta \log(2) - \hat{\gamma}_t + \Delta_t\sqrt{\frac{16L_\ell^2 C_f}{|\mathcal{D}^s|}} + 2C_\ell\sqrt{\frac{\ln(2/\delta)}{2|\mathcal{D}^s|}}. \tag{17}$$

where $\hat{\gamma}_t = \hat{L}_s^A(f) - \hat{L}_s^t(f)$. $\qquad\square$

### C.3 Numerical Experiments

This subsection aims to provide numerical evidence to support the assertion that the value of $\hat{\gamma}_t = \hat{L}_s^A(f) - \hat{L}_s^t(f)$ increases as we increase the value of $t \in \mathbb{N}_0$. Our experimental results, as illustrated in Figure 5, demonstrate that this claim holds true, and that the value of $\hat{\gamma}_t$ becomes positive and remains steady after a few iterations ($t = 10$) of optimizing $W_t$ on the support set $\mathcal{D}^s$.

## D Connection to Meta-Learning

Here we discuss the connection between our Set-SimCLR and meta-learning to clarify why our method can be seen as a unsupervised meta-learning method as follows:

- First, we leverage data augmentation to construct **pseudo-meta-tasks**, where different views of an image belong to **the same pseudo-class**, and **meta-learn** the set-encoder of Set-SimCLR. The set encoder minimizes the distance between positive pairs of set representations and repels negative pairs, where the set representation of the pseudo-class is considered to be a class prototype. In other words, the set encoder **enlarges inter-class distance** so that the set representation of each class eventually leads to a **good initialization of a linear classifier** at meta-test time.

- There are a vast amount of existing meta-learning works proposing to meta-learn the initialization of linear classifiers (Raghu et al., 2019) or amortized neural networks to predict the weight of linear classifiers (Gordon et al., 2019; Iakovleva et al., 2020) by constructing meta-tasks and simulating exact scenarios of meta-test. Similarly we construct the pseudo-meta-tasks and learn the initialization of linear classifiers by simulating meta-test, thus the set encoder of our Set-SimCLR is an indeed **meta-learner**.

- Moreover, Ni et al. (2022) have already highlighted the close relationship between metric-based meta-learning (e.g., Prototypical Networks (Snell et al., 2017)) and contrastive self-supervised learning (Chen et al., 2020a). They claim that we can consider contrastive self-supervised learning as meta-learning since sampling a mini-batch corresponds to sampling a meta-task and contrastive learning with a mini-batch is a $B$-way 1-shot classification problem, where $B$ is mini-batch size. Thus, our feature extractor $f$ which learns through instance and set-level contrastive learning is also a **meta-learner**.

# E    META-SPLIT OF DATASETS

Table 2: The number of classes for meta-split of all datasets.

| Dataset | Meta Train | Meta Valid | Meta Test |
|---|---|---|---|
| Mini-ImageNet | 64 | 16 | 20 |
| Tiny-ImageNet | 100 | 40 | 60 |
| CIFAR100 | 50 | 20 | 30 |
| Aircraft | 50 | 20 | 30 |
| Stanford Cars | 98 | 39 | 59 |
| CUB | 100 | 40 | 60 |

In Table 2, we provide the number of classes for meta-split of all datasets we consider in this paper. Note that we only use meta-test split of Tiny-ImageNet, CIFAR100, Aircraft, Stanford Cars and CUB datasets for the evaluation in Section 4.2.

# F    DETAILS OF BASELINES

In this section, we detail the supervised meta-learning, unsupervised meta-learning and instance-level self-supervised learning baselines. We first introduce two supervised meta-learning approaches which we consider as "*oracles*" and four different unsupervised meta-learning baselines as follows;

**1) MAML (oracle) (Finn et al., 2017)**: Model Agnostic Meta Learning where it learns the initialization of the parameters of the model such that few steps of gradient descent on a support set leads to generalization on a query set. We compare against its performance reported in Hsu et al. (2019).

**2) ProtoNets (oracle) (Snell et al., 2017)**: Euclidean distance-based meta-learning framework. It learns a metric embedding space where we perform prediction by computing a distance between class prototype and instances from query sets. We also compare against it using its performance reported in Hsu et al. (2019).

**3) CACTUs (Hsu et al., 2019)**: Clustering to Automatically Construct Tasks for Unsupervised meta-learning. It automatically constructs tasks by clustering the unsupervised dataset in embedding space learned by ACAI (Berthelot et al., 2019), BiGAN (Donahue et al., 2017), or DeepCluster (Caron et al., 2018). Then it train either MAML or ProtoNets using the cluster indices as pseudo-labels.

**4) UMTRA (Khodadadeh et al., 2019)**: Unsupervised Meta-learning with Tasks constructed by Random sampling and Augmentation. For constructing a K-way 1-shot task, it randomly samples K-way data points from unsupervised dataset and augments each data point. Then MAML is trained on the constructed tasks.

**5) LASIUM (Khodadadeh et al., 2021)**: It trains generative models on the given unlabeled data and sample $N$ different latent vector such that each pair-wise distance is greater than a predefined threshold. Each latent vector is fed into the generative model and decoded to a training instance belonging to distinct class. Then it adds some noise to each latent vector to generate $S$ examples and the generated ones are labeled with the class of the original latent vector. Finally, it trains MAML or ProtoNets using the synthetic $N$-way $S$-shot task.

**6) Meta-GMVAE (Lee et al., 2021)**: Meta-level Gaussian Mixture Variational AutoEncoder. It learns a latent representation by matching set-level amortized variational posterior and task-specific multi-modal prior optimized by EM algorithm.

We then present the four representative self-supervised baselines used in our experiments as follows:

**1) SimCLR (Chen et al., 2020a;b)**: It is a constrative learning framework which learns by maximizing agreement between differently augmented views of the same data example in the latent space.

**2) MOCO (He et al., 2020; Chen et al., 2020c; 2021)**: It builds a dynamic feature dictionary using a queue and momentum encoder and learns to minimize contrastive loss from the dictionary.

**3) BYOL (Grill et al., 2020)**: From pair views of an image, it learns visual representation by matching momentum encoder, which is exponentially moving average of the encoder.

**4) Barlow Twins (Zbontar et al., 2021)**: This method measure the cross-correlation matrix between the feature representations of two different views and learns by making it close to identity matrix.

## G  MASKED AUTOENCODERS

Table 3: The hyperparameters of MAE, which produces the similar number of parameters as ResNet-18 (i.e., VIT: $12, 782, 080$ and ResNet-18: $11, 176, 512$). The name of hyperparameter is based on huggingface transformers library (Wolf et al., 2020).

| Hyperparameters | Value |
|---|---|
| hidden_size | 512 |
| num_hidden_layers | 8 |
| num_attention_heads | 8 |
| intermediate_size | 512 |
| hidden_act | gelu (Hendrycks & Gimpel, 2016) |
| hidden_drop_prob | 0.0 |
| initializer_range | 0.02 |
| layer_norm_eps | $10^{-12}$ |
| is_encoder_decoder | False |
| image_size | 84 |
| patch_size | 6 |
| num_channels | 3 |
| qkv_bias | True |
| decoder_num_attention_heads | 8 |
| decoder_hidden_size | 128 |
| decoder_num_hidden_layers | 3 |
| decoder_intermediate_size | 128 |
| mask_ratio | 0.75 |
| norm_pix_loss | True |

Table 4: Results for 5-way $S$-shot classification on Mini-ImageNet. We report mean and standard deviation of accuracy evaluated on 1000 episodes with 5 different runs, except for MAE. For MAE, we report mean for accuracy for one run.

| Method | Base Encoder | 1-shot | 5-shot | 20-shot | 50-shot |
|---|---|---|---|---|---|
| SimCLR | | $46.23_{\pm0.31}$ | $67.08_{\pm0.26}$ | $76.51_{\pm0.23}$ | $80.14_{\pm0.45}$ |
| MOCO | | $43.96_{\pm0.35}$ | $62.64_{\pm0.14}$ | $72.21_{\pm0.35}$ | $78.02_{\pm0.20}$ |
| BYOL | ResNet-18 | $45.59_{\pm1.57}$ | $64.19_{\pm1.29}$ | $73.97_{\pm1.26}$ | $76.55_{\pm1.63}$ |
| Barlow Twins | | $45.12_{\pm0.19}$ | $63.44_{\pm0.27}$ | $72.13_{\pm0.27}$ | $75.92_{\pm0.25}$ |
| **Set-SimCLR (ours)** | | $\mathbf{53.54}_{\pm0.66}$ | $\mathbf{69.79}_{\pm0.28}$ | $\mathbf{78.53}_{\pm0.26}$ | $\mathbf{82.10}_{\pm0.47}$ |
| MAE with lr = 0.002 | | 34.47 | 46.40 | 56.62 | 62.94 |
| MAE with lr = 0.001 | VIT in Table 3 | 31.46 | 42.55 | 53.84 | 59.91 |
| MAE with lr = 0.0005 | | 32.38 | 45.83 | 58.61 | 65.65 |

MAE (He et al., 2022) is a recent self-supervised learning method based on masked auto-encoding objective. We tried to use MAE as a baseline, and the experimental setups are as follows. It assumes VIT (Dosovitskiy et al., 2021) as a base encoder, therefore, we use the hyperparameters in Table 3 which produce the similar amount of parameters as ResNet-18 (i.e., VIT: 12782080 and ResNet-18: 11176512). We use huggingface transformers library (Wolf et al., 2020) for implementation. Following the original implementation of MAE, we optimize MAE using AdamW (Loshchilov & Hutter, 2019) with 0.05 for 400 epochs. The mini-batch size is set to 512. We search the adequate learning rate in 0.002, 0.001, 0.0005 using meta-validation split. We use cosine learning rate scheduler with 40 warm-up epochs. We use `ResizedCrop`, `HorizontalFlip` for augmentations. In Table 4 shows the mean accuracy of ours, self-supervised learning baselines and MAE on the Mini-ImageNet 5-way few-shot classification tasks. We found that MAE fails to achieve comparable performance in our UML setting, therefore, we exclude it in our main text.

## H    IMPLEMENTATION DETAILS OF SECTION 4.1

Table 5: The architecture of Conv5 used as a base encoder $f$ for the experiments in Sec 4.1.

| Output Size | Layers |
|---|---|
| $3 \times 84 \times 84$ | Input Image |
| $64 \times 42 \times 42$ | $\texttt{Conv2d}(3 \times 3, \text{stride} = 1, \text{pad} = 1), \texttt{BatchNorm2D}, \texttt{ReLU}, \texttt{Maxpool}(2 \times 2, \text{stride} = 2)$ |
| $64 \times 21 \times 21$ | $\texttt{Conv2d}(3 \times 3, \text{stride} = 1, \text{pad} = 1), \texttt{BatchNorm2D}, \texttt{ReLU}, \texttt{Maxpool}(2 \times 2, \text{stride} = 2)$ |
| $64 \times 10 \times 10$ | $\texttt{Conv2d}(3 \times 3, \text{stride} = 1, \text{pad} = 1), \texttt{BatchNorm2D}, \texttt{ReLU}, \texttt{Maxpool}(2 \times 2, \text{stride} = 2)$ |
| $64 \times 5 \times 5$ | $\texttt{Conv2d}(3 \times 3, \text{stride} = 1, \text{pad} = 1), \texttt{BatchNorm2D}, \texttt{ReLU}, \texttt{Maxpool}(2 \times 2, \text{stride} = 2)$ |
| $64 \times 2 \times 2$ | $\texttt{Conv2d}(3 \times 3, \text{stride} = 1, \text{pad} = 1), \texttt{BatchNorm2D}, \texttt{ReLU}, \texttt{Maxpool}(2 \times 2, \text{stride} = 2)$ |
| 256 | $\texttt{Flatten}$ |

Table 6: The architecture of set encoder $\varphi$ used for the experiments in Sec 4.1.

| Output Size | Layers |
|---|---|
| $M \times 256$ | $M$ Input Features |
| $M \times 256$ | $\texttt{TransformerEncoder}(d_{\text{model}} = 256, d_{\text{ff}} = 256, \text{num\_heads} = 4, \texttt{ReLU})$ |
| 1024 | $\texttt{concat}\,(\texttt{mean}(\cdot); \texttt{std}(\cdot); \texttt{max}(\cdot); \texttt{min}(\cdot))$ |
| 256 | $\texttt{Linear}(1024, 256), \texttt{ReLU}$ |
| 256 | $\texttt{Linear}(256, 256), \texttt{ReLU}$ |
| 256 | $\texttt{Linear}(256, 256)$ |

Table 7: The architecture of head $g$ used for the experiments in Sec 4.1.

| Output Size | Layers |
|---|---|
| 256 | Input Feature |
| 256 | $\texttt{Linear}(256, 256), \texttt{BatchNorm1d}, \texttt{LeakyReLU}$ |
| 64 | $\texttt{Linear}(256, 64)$ |

We provide pytorch-like architecture implementations of base encoder $f$, set encoder $\varphi$ and head $g$ in Table 5, 6 and 7, respectively. We follow SimCLR (Chen et al., 2020a;b) for random augmentation, which is detailed in Appendix J. We apply the composed augmentations to 64 mini-batch images eight times (i.e., $M = 64, V = 8$), resulting in 4 elements in each set. We optimize the base encoder, set encoder and head network for 400 epochs using Adam optimizer (Kingma & Ba, 2015) with default settings (i.e., $\beta_1 = 0.9$ and $\beta_2 = 0.999$). We use constant learning rate of $0.001$. For downstream tasks, we use scikit-learn (Pedregosa et al., 2011) package with default settings to optimize a linear classifier.

## I    IMPLEMENTATION DETAILS OF SECTION 4.2

Table 8: The architecture of set encoder $\varphi$ used for the experiments in Sec 4.2.

| Output Size | Layers |
|---|---|
| $M \times 512$ | $M$ Input Features |
| $M \times 512$ | $\texttt{TransformerEncoder}(d_{\text{model}} = 512, d_{\text{ff}} = 512, \text{num\_heads} = 4, \texttt{ReLU})$ |
| 2048 | $\texttt{concat}\,(\texttt{mean}(\cdot); \texttt{std}(\cdot); \texttt{max}(\cdot); \texttt{min}(\cdot))$ |
| 512 | $\texttt{Linear}(2048, 512), \texttt{ReLU}$ |
| 512 | $\texttt{Linear}(512, 512), \texttt{ReLU}$ |
| 512 | $\texttt{Linear}(512, 512)$ |

For the base encoder $f$, we use ResNet-18 architecture. Please see the original paper (He et al., 2016) for implementation details. We provide pytorch-like architecture implementations of set encoder $\varphi$ and head $g$ in Table 8 and 9, respectively. For a fair comparison, we use the same architecture of head network $g$ in Table 9, for all self-supervised learning methods except for MOCO. MOCO does not use the head as firstly proposed in the original paper. We use the same random augmentations described

Table 9: The architecture of head $g$ used for the experiments in Sec 4.2.

| Output Size | Layers |
|---|---|
| 512 | Input Feature |
| 512 | `Linear`(512, 512), `BatchNorm1d`, `LeakyReLU` |
| 128 | `Linear`(512, 128) |

Table 10: The selected learning rate of each method.

| Method | Learning Rate |
|---|---|
| SimCLR | 0.001 |
| MOCO | 0.001 |
| BYOL | 0.0005 |
| Barlow Twins | 0.001 |
| Set-SimCLR (ours) | 0.0005 |

in Appendix J. For our method Set-SimCLR, we apply the augmentations 8 times to the mini-batch of 64 images (i.e., $M = 64, V = 8$), resulting in 4 elements in each set, while performing the same augmentation twice on the mini-batch of 256 images (i.e., $M = 256, V = 2$) for the other baselines. For all the methods, we optimize the models for 400 epochs using Adam optimizer (Kingma & Ba, 2015) with default settings (i.e., $\beta_1 = 0.9$ and $\beta_2 = 0.999$). We do not use learning rate scheduling which is not effective for any methods in our experiments. We search for an adequate learning rate in 0.001, 0.0005, 0.0001 for baselines and ours using a meta-validation split. We provide the selected learning rate of each method in Table 10. We use scikit-learn (Pedregosa et al., 2011) package with default settings to optimize classifiers for downstream tasks.

## J  RANDOM AUGMENTATION

Table 11: The application probability and hyperparameters of each augmentation.

| Augmentation | Probability | Hyperparameters |
|---|---|---|
| `ResizedCrop` | 1.0 | size $= (84, 84)$, scale $= (0.08, 1.0)$, ratio $= (0.75, 1.3...)$ |
| `HorizontalFlip` | 0.5 | N/A |
| `ColorJitter` | 0.8 | brightness $= 0.8$, contrast $= 0.8$, saturation $= 0.8$, hue $= 0.2$ |
| `GrayScale` | 0.2 | N/A |
| `GaussianBlur` | 0.5 | kernel_size $= (85, 85)$, $\sigma \sim U(0.1, 2.0)$ |

For random augmentation, we compose `ResizedCrop`, `HorizontalFlip`, `ColorJitter`, `GrayScale` and `GaussianBlur`. The application probability and hyperparameters of each augmentation is shown in Table 11. Note that we perform `ResizedCrop` on a larger resolution of $224 \times 224$ images than the resolution of $84 \times 84$ images we target, which is found to be more effective. We implement the augmentation using Kornia framework (Riba et al., 2020), which allows a faster augmentations on GPU.

## K  WALL-CLOCK TIME FOR SSL METHODS

Table 12: We report the average wall-clock time to train SSL methods for 400 training epochs.

| Method | Wall-Clock Time for 400 epochs |
|---|---|
| SimCLR | 9h 38m 54s |
| MOCO | 10h 36m 36s |
| BYOL | 10h 40m 34s |
| Barlow Twins | 10h 28m 8s |
| SetSimCLR | 22h 45m 3s |

## L  Full Tables for Figure 2

Table 13: 5-way 5-shot classification results on Aircraft, Stanford Cars, CIFAR100, CUB, Mini-ImageNet and Tiny-ImageNet datasets. The base encoder is ResNet-18. We report the mean and standard deviation of 5 runs with different random seeds.

| Method | Mini | Tiny | CIFAR100 | Aircraft | Cars | CUB |
|---|---|---|---|---|---|---|
| *Training from Scratch* | 34.22 $\pm$0.45 | 34.11 $\pm$0.44 | 42.36 $\pm$0.56 | 36.82 $\pm$0.49 | 29.29 $\pm$0.39 | 33.34 $\pm$0.42 |
| SimCLR | 67.08 $\pm$0.26 | 66.06 $\pm$0.34 | 64.27 $\pm$1.35 | 46.36 $\pm$0.11 | 37.05 $\pm$0.12 | 47.30 $\pm$0.30 |
| MOCO | 62.64 $\pm$0.14 | 60.67 $\pm$0.41 | 60.75 $\pm$0.83 | 46.81 $\pm$0.54 | 38.33 $\pm$0.57 | 47.02 $\pm$0.13 |
| BYOL | 64.19 $\pm$1.52 | 63.83 $\pm$1.31 | 65.95 $\pm$1.73 | 44.29 $\pm$0.49 | 35.90 $\pm$0.92 | 45.95 $\pm$1.03 |
| Barlow Twins | 63.44 $\pm$0.27 | 62.20 $\pm$0.26 | 63.25 $\pm$0.52 | 46.05 $\pm$0.45 | 34.70 $\pm$0.23 | 44.73 $\pm$0.25 |
| **Set-SimCLR (ours)** | **69.79** $\pm$0.28 | **67.27** $\pm$0.18 | **66.85** $\pm$1.76 | **47.49** $\pm$0.39 | **38.50** $\pm$0.37 | **49.00** $\pm$0.31 |

Table 14: 5-way 1, 5, 20, 50-shot classification results on Mini-ImageNet dataset. The base encoder is ResNet-18. We report the mean and standard deviation of 5 runs with different random seeds.

| Method | 1-shot | 5-shot | 20-shot | 50-shot |
|---|---|---|---|---|
| *Training from Scratch* | 24.86 $\pm$0.36 | 34.22 $\pm$0.45 | 45.60 $\pm$0.48 | 53.01 $\pm$0.49 |
| SimCLR | 46.23 $\pm$0.31 | 67.08 $\pm$0.26 | 76.51 $\pm$0.23 | 80.14 $\pm$0.45 |
| MOCO | 43.63 $\pm$0.35 | 62.64 $\pm$0.14 | 72.21 $\pm$0.35 | 78.02 $\pm$0.20 |
| BYOL | 45.59 $\pm$1.57 | 64.19 $\pm$1.29 | 73.97 $\pm$1.26 | 76.55 $\pm$1.63 |
| Barlow Twins | 45.12 $\pm$0.19 | 63.44 $\pm$0.27 | 72.13 $\pm$0.27 | 75.92 $\pm$0.25 |
| **Set-SimCLR (ours)** | **53.54** $\pm$0.66 | **69.79** $\pm$0.28 | **78.53** $\pm$0.26 | **82.10** $\pm$0.47 |

## M  More Ablation Studies

Table 15: 5-way $N$-shot classification results of SimCLR, Set-SimCLR without set representation at meta test, and original Set-SimCLR on Mini-ImageNet. The base encoder is ResNet-18. We report the mean and standard deviation of 5 runs with different random seeds.

| Method | Set | 1-shot | 2-shot | 5-shot | 20-shot | 50-shot |
|---|---|---|---|---|---|---|
| SimCLR | ✗ | 46.23 $\pm$0.31 | 55.58 $\pm$0.39 | 67.08 $\pm$0.26 | 76.51 $\pm$0.23 | 80.14 $\pm$0.45 |
| Set-SimCLR | ✗ | 49.34 $\pm$0.57 | 59.10 $\pm$0.28 | 69.03 $\pm$0.54 | 77.95 $\pm$0.20 | 81.68 $\pm$0.34 |
| Set-SimCLR | ✓ | **53.54** $\pm$0.66 | **60.87** $\pm$0.24 | **69.79** $\pm$0.28 | **78.53** $\pm$0.26 | **82.10** $\pm$0.47 |

To understand the performance gain of Set-SimCLR step-by-step, we conduct an additional ablation study by comparing the full model Set-SimCLR against SimCLR, and Set-SimCLR without the initialization of classifier weight using set representations. Table 15 shows that Set-SimCLR without set initialization, improves the generalization performance of the model trained with only SimCLR loss by $1.44\% \sim 3.54\%$. Thus, the performance gain is a consequence of introducing set-level loss. If we leverage learned set representation to initialize the weight $W$ (Set-SimCLR with set), we can further boost the performance of the model Set-SimCLR without set by $0.42\% \sim 4.2\%$. We further observe the performance gain becomes larger for fewer shots. Therefore, learning a set representation with our proposed set-level loss is crucial for better generalization performance.

Table 16: 5-way $N$-shot Mini-ImageNet classification results Set-SimCLR with the parameters $W_t$ at different optimization steps ($t = 0, 20, 100$). The base encoder is ResNet-18. We report the mean and standard deviation of 5 runs with different random seeds.

| Method | $t$ | 1-shot | 5-shot | 20-shot | 50-shot |
|---|---|---|---|---|---|
| SimCLR | 0 | 47.91 $\pm$1.53 | 57.34 $\pm$1.24 | 59.78 $\pm$1.63 | 60.17 $\pm$1.51 |
| Set-SimCLR | 20 | 52.94 $\pm$0.42 | 69.22 $\pm$0.24 | 78.27 $\pm$0.24 | 81.90 $\pm$0.22 |
| Set-SimCLR | 100 | **53.54** $\pm$0.66 | **69.79** $\pm$0.28 | **78.53** $\pm$0.26 | **82.10** $\pm$0.47 |

In Table 16, we provide the performance on 5-way N-shot Mini-ImageNet with the parameters $W_t$ at different optimization steps ($t = 0, 20, 100$) for fine-tuning. Though Set-SimCLR performs not that good at $t = 0$, it rapidly adapts to support sets to reach near the best accuracy at $t = 20$.

