# OpenReview forum: "Self-Supervised Set Representation Learning for Unsupervised Meta-Learning"
_ICLR.cc/2023/Conference — ICLR 2023 poster_

### Official Review · Reviewer_DFwo · 2022-10-22

**Confidence:** 3
**Correctness:** 3
**Technical Novelty And Significance:** 2
**Empirical Novelty And Significance:** 1
**Recommendation:** 5

**Clarity, Quality, Novelty And Reproducibility:**

The paper is clearly written.  The main novelty of the work lies in extending SimCLR and as such the originality is marginal. The work should be highly reproducible but there is no code available. Authors are advised to release the code.

**Strength And Weaknesses:**

Strengths:
- The paper is clearly written and well organized.
- The idea of combining set representation and SimCLR is neat and effective.
- The experiments show improvement compared to existing unsupervised meta-learning and self-supervised learning methods.

Weaknesses:
- The main novelty of the method lies in extending SimCLR to set representation which seems marginal. Also, it is unclear why the authors propose this method for unsupervised meta-learning since they do not propose a new meta-learning framework (nothing is meta-learned here) but rather self-supervised learning framework. It would be more natural to claim that Set-SIMCLR is a new self-supervised learning method and then demonstrate that it can be used for few-shot learning and it can achieve performance competitive to unsupervised meta-learning methods.
- The φ in linear evaluation for downstream tasks is unclear.The authors state that base encoder f is fixed during fine-tuning, and the linear classifier is updated. How about the set encoder which is the most important part of the method?
- In the chapter Theoretical Motivation, one of the main goals is to study how the set representation can potentially improve the final performance. In the last paragraph of this chapter, the author states that Theorem 1 provides a theoretical motivation on the use of the set representation (to increase γ at t = 0). However, it is unclear how the set representation increases γ at t = 0.
- In the Training Budget Analysis, the author trains the baseline for 800 epochs, which is twice larger than before. The goal is to train with a time equal to the training time of Set-SimCLR, however, we do not know the training time of Set-SimCLR. For example, if the training time of Set-SimCLR is ten times larger than the baselines, training the baselines with twice larger time is not enough.
- In the experiments in Table 1, authors use Conv4 or Conv5 as the backbone. Does it mean that different backbone is used for different baselines? Additionally, how would the results look like with the ResNet-18 backbone which is used in later experiments?
- Why is the performance so variable with the different number set cardinality on the 1-shot learning task? If the set representation is indeed helpful, shouldn’t larger cardinality consistently improve performance (but of course lead to larger computational time)?




**Summary Of The Paper:**

This paper proposes a new approach for unsupervised meta-learning problem. The key idea of the approach, named  Set-SimCLR, is to extend the SimCLR framework with the set representation. The proposed approach augments the set of images and then learns set representation using a set transformer.  Additionally, the authors theoretically study how the set representation potentially improves few-shot classification performance. The method is compared to unsupervised meta-learning and self-supervised learning techniques on benchmark datasets.


**Summary Of The Review:**

The idea of combining set representation and SimCLR is clear and simple, and it works well in experiments with several datasets. Also, the paper is well organized. However, the novelty is marginal and some details remain unclear or are not convincing.

---

### Official Review · Reviewer_97Aw · 2022-10-23

**Confidence:** 3
**Correctness:** 3
**Technical Novelty And Significance:** 3
**Empirical Novelty And Significance:** 3
**Recommendation:** 6

**Clarity, Quality, Novelty And Reproducibility:**

The paper is well written and easy to follow. The paper is technically sound and novel to me.



**Strength And Weaknesses:**

Strength: The motivation and the insight of this submission are clear. The authors give a comprehensive and sufficient introduction to the background of UML and SSL. Viewing the UML as the set-level problem, the authors  exploits the effectiveness of self-supervised learning into the set representation learning. Then the authos use the set representation from the support set as the initialization W0 and optimize W with supervised loss. Besides, the authors provide theoretical motivations on several aspects of their algorithmic design and rich experimental results on several downstream few-shot classification tasks.


Weaknesses: Do the authors classify the meta test tasks using W0 ?  I am curious about what classification performance could be achieved only with the initial parameters? It is clear that  Set-SimCLR outperforms existing UML methods. Compared with SSL methods, Set-SimCLR still performs better. Could we attribute this result to the set representation learning, i.e., introducing the set-level loss?  I wonder whether the authors can provide more discussions from this view. Besides, the effectiveness of their proposed method on transfer learning scenarios should be stressed i section 4.2.   It would be better if the authors present the Algorithm about Set-SimCLR.


**Summary Of The Paper:**

This work develops a self-supervised set representation learning framework for unsupervised meta-learning (UML), which learns instance and set representations simultaneously for downstream tasks. The authors theoretically analyze how Set-SimCLR can potentially improve the generalization performance at the meta-test and empirically validate its effectiveness on various benchmark datasets.

**Summary Of The Review:**

The paper is probably publishable, but should be reviewed again before it is accepted.

---

### Official Review · Reviewer_vy1j · 2022-10-24

**Confidence:** 4
**Correctness:** 3
**Technical Novelty And Significance:** 3
**Empirical Novelty And Significance:** 3
**Recommendation:** 6

**Clarity, Quality, Novelty And Reproducibility:**

Overall, the paper is well-written and the method appears to be a non-trivial combination of SSL and set representations for UML.  However, I do have some concerns, as I wrote about in my strength/weaknesses above.

**Strength And Weaknesses:**

Strengths:
1) The empirical study is well-done and makes a strong case that Set-SimCLR would outperform other UML methods/instance level SSL methods.  Error bars give me confidence that the outperformance is something reproducible.
2) The method is simple and seems easy to implement.
3) The paper is mostly well-written and easy to understand.

Weaknesses (please correct me if I misunderstood anything):
1) If we are freezing the encoder f, then the supervised learning loss at meta-test is just logistic regression.  Why would initializing W (using the set representations) make a difference in that case?  This is a convex optimization problem.  I would like to see an experiment that shows what happens if we do not initialize W as suggested in Set-SimCLR.  Note this is different from just doing SimCLR, since the pretraining loss is different.  I would also like some explanation on why initializing W matters at a conceptual level.
2) The statement of Theorem 1 is confusing to me.  If we expand \gamma, isn't this saying the cL^{\text{SimCLR}} - \zeta - L^A_s(f) \geq 0? Which doesn't seem meaningful to me.  Let me know if I've made a mistake here.
3) I would like an additional comparison to a baseline where we do no pretraining and just train from scratch on each of the meta-test tasks.  The goal here would be to make sure that our pre-trained representation is useful in the first place.

Typo:
"The second interpretation is obtained by setting the probability measures to be the probability measures,"

**Summary Of The Paper:**

This paper proposes a new method, Set-SimCLR, which is a version of SimCLR that produces set representations for use in unsupervised meta learning.  Set-SimCLR combines two ideas: (1) SimCLR, which is an instance-level self-supervised learning method and has been shown to produce useful representations in downstream supervised learning tasks (and therefore might also work well in the UML) (2) set representations have been shown to be useful in meta-learning, where the set are all examples of the same class.  Some theoretical motivation is given for Set-SimCLR, and empirical evidence shows that Set-SimCLR performs better than other UML/SSL methods.

**Summary Of The Review:**

The empirical case for Set-SimCLR, the method in this paper, was fairly convincingly made for me.  However, I have some reservations conceptually and theoretically, as stated in the weaknesses.  If they are adequately addressed, then I am willing to revise my rating.

============
Rebuttal: my concerns about the paper have been addressed adequately.  I revise my score to a 6.  I recommend acceptance.
============

---

> ### Comment · Reviewer_vy1j · 2022-11-22
> **Thanks for the rebuttal**
>
> My concerns have been addressed, and the experimental results are compelling.  I recommend acceptance.

---

### Official Review · Reviewer_KksJ · 2022-10-30

**Confidence:** 4
**Correctness:** 3
**Technical Novelty And Significance:** 2
**Empirical Novelty And Significance:** 2
**Recommendation:** 5

**Clarity, Quality, Novelty And Reproducibility:**

The paper is well-written and easy to understand.

In terms of novelty, the paper lifts simCLR to set-based simCLR which is modest but useful.

Given the details on architecture and hyper-parameters, I find the paper reproducible.

**Strength And Weaknesses:**



This paper overviews some of the important previous works on metric learning-based meta-learning. An observation of set-based representation for meta-learning in the previous works on meta-learning based on metric learning motivates this work on set-based self-supervised learning, which is convincing. The paper uplifts the existing simCLR to satisfy the pairwise set-based positive and negative constraints. The method is simple and easy to understand. The related works are also adequately present which is also a strength of this paper.

Extensive experiments are performed on various benchmarks.  Studies of hyper-parameters is also adequate. The experimental results show the effectiveness of the paper.


The only concern I have is on the outcome of the study on the cardinality of the set. In Figure 4(c), the study shows that the method is insensitive to the cardinality of the set. I believe when the cardinality is 2, the method is equivalent to simCLR. If the method is insensitive to cardinality, then from where, do we obtain the performance gain?


**Summary Of The Paper:**

This paper presents set-SimCLR, a set representation-based self-supervised learning method for unsupervised meta-learning. To this end, the paper extracts a representation of images with various augmentations from a base model. Partition the set of features from augmented images into two and aggregate them to obtain a positive pair of set-based representations. The other set representation in a batch makes negative pairs.  The parameters are learned to minimize both the set-based and instance-based contrastive loss. Extensive experiments on six different benchmarks are performed and compared with several existing methods.




**Summary Of The Review:**

The outcome of the cardinality study is one of my main concerns. I hope the authors will address it convincingly on rebuttal.

---

> ### Author Response · Authors · 2022-12-02
> **A Gentle Reminder**
>
> Dear Reviewer KksJ,
>
> We sincerely appreciate your time and effort in reviewing our paper. During the discussion period, we have made every effort to faithfully address all your comments in the responses, by providing clarifications on your questions and revising the paper. We strongly believe that the paper is now significantly strengthened thanks to your constructive comments. Thus, we politely ask you to go over our responses and reconsider your rating on our work. Please let us know if you have any further questions.
>
> Best regards, Authors

---

### Decision · Program_Chairs · 2023-01-20

**Decision:**

Accept: poster

**Justification For Why Not Higher Score:**

Theoretical statement may be confusing or misleading.

**Justification For Why Not Lower Score:**

Novel SSL framework and good performance on downstream tasks.

**Metareview: Summary, Strengths And Weaknesses:**

Paper studies Unsupervised Meta-Learning (UML) for the challenging task of Few-shot Classification (FSC). This done through a new self-supervised pre-training framework called Set-SimCLR. Set-SimCLR augments SimCLR framework by learning "set representations" for "sets of augmentations of an instance" in addition to "augmented-instance representations". Further these two types of representations are semantically linked through a shared encoder head.

Next, the authors use  "instance representations" as features  for the downstream Few-shot k-way Classification task. The task is solved using a linear classifier which is initialized with the "set representations" of few instance of different classes. Finally, the linear classifier is further tuned while keeping the features fixed. The framework is empirically shown to achieve much better performance than baseline UML method for FSC at the cost of increased computations. Reviewers identified lack of clarity in-terms of method, experiments and theory and authors fixed many of these issues.

The theoretical motivation given for this work may be inadequate. Specifically, it is shown that good initialization for the few-shot linear classifier is useful when there is labeled data scarcity. However, this is sort of obvious and known in many forms in past literature. What is not proven is that the learned "set representations" are good initializers. Therefore the following statement from the authors in the discussion may be misleading to readers
> "Lastly, we theoretically motivate the importance of learning set representation and initializing the weights of classifiers with set representations."

Theorem also proves that minimizing original SimCLR loss improves downstream performance. It is not shown if the bound is tight and not vacuous. For example, can the term $\hat{\gamma}_t$ be negative? If yes, then it might better to initialize with average pooling $W_0 = A$. This would be a good future experiment. Theory doesn't take into account the the "set representation" learning.

Nit: Typo in row 1 of Table 16, Method should be Set-SimCLR at $t=0$.

**Note From Pc:**

if the above contains the word "oral" or "spotlight" please see: "oral" presentation means -> notable-top-5% and "spotlight" means -> notable-top-25%. As stated in our emails, we are disassociating presentation type from AC recommendations